# Bilevel Network Learning via Hierarchically Structured Sparsity

Jiayi Fan[1]*, Jingyuan Yang[1]*, Shuangge Ma[2], Mengyun Wu[1]†

[1]School of Statistics and Data Science, Shanghai University of Finance and Economics
[2]Department of Biostatistics, Yale School of Public Health
`fanjiayi@stu.sufe.edu.cn`, `yang.jingyuan@stu.sufe.edu.cn`,
`shuangge.ma@yale.edu`, `wu.mengyun@mail.shufe.edu.cn`

## Abstract

Accurate network estimation serves as the cornerstone for understanding complex systems across scientific domains, from decoding gene regulatory networks in systems biology to identifying social relationship patterns in computational sociology. Modern applications demand methods that simultaneously address two critical challenges: capturing nonlinear dependencies between variables and reconstructing inherent hierarchical structures where higher-level entities coordinate lower-level components (e.g., functional pathways organizing gene clusters). Traditional Gaussian graphical models fundamentally fail in these aspects due to their restrictive linear assumptions and flat network representations. We propose NNBLNet, a neural network-based learning framework for bi-level network inference. The core innovation lies in hierarchical selection layers that enforce structural consistency between high-level coordinator groups and their constituent low-level connections via adaptive sparsity constraints. This architecture is integrated with a compositional neural network architecture that learn cross-level association patterns through constrained nonlinear transformations, explicitly preserving hierarchical dependencies while overcoming the representational limitations of linear methods. Crucially, we establish formal theoretical guarantees for the consistent recovery of both high-level connections and their internal low-level structures under general statistical regimes. Extensive validation demonstrates NNBLNet's effectiveness across synthetic and real-world scenarios, achieving superior F1 scores compared to competitive methods and particularly beneficial for complex systems analysis through its interpretable bi-level structure discovery.

## 1 Introduction

Network estimation is a fundamental task across many disciplines—such as genetics, finance, and social science—where uncovering the structure of dependencies among variables can yield critical insights into the underlying mechanisms of complex systems [8, 38, 44]. For instance, in genetics, gene regulatory networks can be reconstructed from analyzing multi-patient omics datasets to identify statistically significant edges between genes. In social science, co-authorship networks connect researchers through jointly published papers, with edge weights reflecting collaboration intensity, and the resulting network reveals meaningful patterns of scholarly connection and knowledge flow.

A key characteristic shared across these domains is that variables can often be naturally organized into groups. Such group information is readily available as direct labels in practice: for instance, most genes have well-annotated pathway information in databases like KEGG, which naturally defines

---

*These authors contributed equally.
†Corresponding author. Email: `wu.mengyun@mail.shufe.edu.cn`

functional groupings, while public social datasets commonly include demographic or professional categories such as gender or research field that provide meaningful group divisions. Under such a grouped organization, connections may occur both within these groups—reflecting functional or structural coherence—and between groups—reflecting higher-order dependencies or cross-functional regulation [26, 6]. Crucially, these systems exhibit hierarchical dependency architectures: dependencies among low-level variables often emerge only when their parent groups share systemic interdependencies. For instance, in genomics, genes within a pathway rarely interact with genes in unrelated pathways unless cross-pathway regulatory mechanisms exist [14, 10]. This hierarchical structure poses significant modeling challenges, as conventional network estimation methods that ignore group-level dependencies often fail to capture emergent system behaviors, yielding fragmentary results that lack systematic interpretability. This motivates bi-level network estimation, which aims to recover hierarchically structured dependencies at both the group and variable levels.

Gaussian Graphical Models (GGMs) are widely used for learning network structures by estimating the inverse covariance matrix under multivariate Gaussian assumptions due to their ability to capture conditional dependencies between variables [39]. Furthermore, a series of methods reformulate the GGM estimation problem into a set of sparse linear regression models, often offering greater computational efficiency [24]. Extensions to bi-level GGMs have also been proposed to capture grouped variable associations [29]. However, GGMs are restricted to modeling linear relationships and rely heavily on Gaussianity, limiting their applicability in real-world data with nonlinear or non-Gaussian structure. To address this gap, a number of model-free or nonparametric approaches have been proposed [13, 36, 32]. While these methods alleviate distributional assumptions, they often struggle to model highly complex interdependencies in modern high-dimensional systems, where traditional kernel-based or graphical techniques lack the representational capacity to capture intricate hierarchical patterns. Neural network (NN) has demonstrated superior performance in capturing nonlinear relationships compared to conventional nonparametric methods [18]. Nevertheless, neural network-based bi-level network estimation remains unexplored.

We propose a neural network–based framework for bi-level network estimation to address the challenge of modeling complex, hierarchical dependencies in grouped variables. This work pioneers the use of neural networks for bi-level network inference, effectively capturing nonlinear interdependencies both within and across groups, beyond the capacity of traditional linear models. The key contributions of our work are:

- Bi-level Network Estimation: We propose a structured estimation framework that recovers hierarchical network architectures through dual-layer selection mechanisms, capturing bi-level dependencies between variables by incorporating group information and identifying group dependencies simultaneously.

- Hierarchical Nonlinear Architecture: We integrate compositional neural network architectures to model cross-level dependencies via constrained nonlinear transformations, explicitly preserving hierarchical structures while overcoming the representational constraints of linear methods.

- Theoretical Guarantees: We formally introduce the notion of *bi-level selection consistency* and establish a rigorous selection consistency result under high-dimensional regimes, ensuring the method to reliably recover the true network structure at both levels.

- Empirical Validation: We validate our method through comprehensive experiments on both synthetic and real-world datasets. The results highlight the advantages of our approach in accurately estimating complex network structures, especially in scenarios where traditional GGM-based methods fail due to nonlinearity or distributional misspecification.

## 1.1 Notation

Consider $n$ independent observations with $p$ variables partitioned into $L$ predefined groups (Figure 1 A), where the $l$-th group contains $p_l$ variables such that $\sum_{l=1}^{L} p_l = p$. For the $i$-th observation, define the continuous measurement vector as $\boldsymbol{x}_i = (x_{i1}, \ldots, x_{ip})^\top$. The group membership of the $j$-th variable is specified by $C_j \in \{1, \ldots, L\}$. We further define two derived quantities: $\boldsymbol{x}_{i,-j} = (x_{i,1}, \ldots, x_{i,j-1}, x_{i,j+1}, \ldots, x_{i,p})^\top$ representing the $i$-th observation with the $j$-th variable excluded and $\tilde{\boldsymbol{x}}_{il}$ denoting the subvector containing all variables from group $l$ in the $i$-th observation. We define the operator $\odot$ as a block-wise product between a vector and

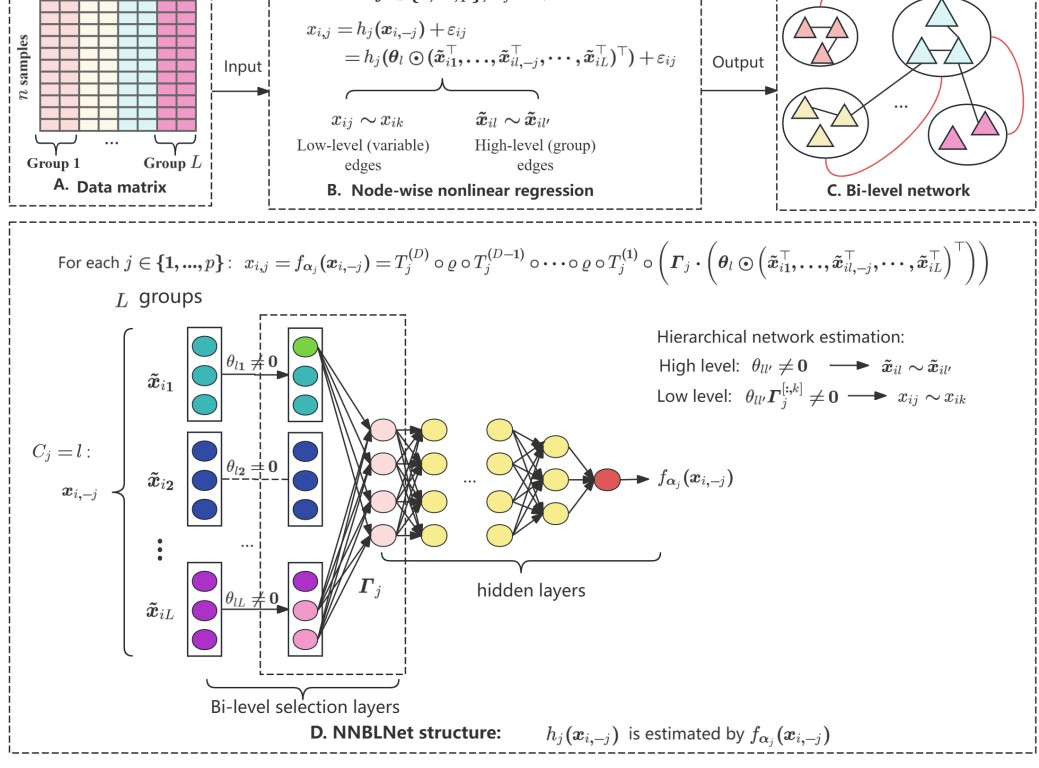

Figure 1: Workflow of the proposed bi-level network estimation. (A) Input $\boldsymbol{x}_i = (x_{i,1}, \cdots, x_{i,p})^\top$'s: $n$ observations with $p$ variables partitioned into $L$ groups. (B) Node-wise nonlinear regression for modeling the relationships between the $j$th variable $x_{i,j}$ and the others $\boldsymbol{x}_{i,-j}$, as well as those between groups $\tilde{\boldsymbol{x}}_{i,l}$ and $\tilde{\boldsymbol{x}}_{i,l'}$. Here, $C_j = l$ indicates that the $j$th variable belongs to the $l$th group. (C) Output: Bi-level network, including both low-level edges and high-level edges. (D) Network structure of the proposed NNBLNet, including the two hierarchical selection layers and multiple hidden layers for accommodating the nonlinear relationships among both low-level variables and high-level groups.

a partitioned vector (or matrix). Specifically, let $\boldsymbol{A} = (a_1, \ldots, a_m)^\top \in \mathbb{R}^{m \times 1}$ be a vector and $\boldsymbol{B} = \left( \left( \boldsymbol{B}^{(1)} \right)^\top, \ldots, \left( \boldsymbol{B}^{(m)} \right)^\top \right)^\top \in \mathbb{R}^{n \times N}$ be a matrix that can be decomposed into $m$ contiguous blocks, i.e., $\boldsymbol{B}^{(l)} \in \mathbb{R}^{n_l \times N}$ for $l = 1, \ldots, m$ with $\sum_{l=1}^{m} n_l = n$. The block-wise product $\boldsymbol{A} \odot \boldsymbol{B} \in \mathbb{R}^{n \times N}$ is defined as $\boldsymbol{A} \odot \boldsymbol{B} := \left( a_1 \left( \boldsymbol{B}^{(1)} \right)^\top, a_2 \left( \boldsymbol{B}^{(2)} \right)^\top, \ldots, a_m \left( \boldsymbol{B}^{(m)} \right)^\top \right)^\top$. That is, each element $a_l$ scales all entries in the corresponding block $\boldsymbol{B}^{(l)}$.

## 2 Neural Network-based Bi-level Network Estimation (NNBLNet)

Traditional methods, such as GGMs, typically handle linear conditional dependencies captured via precision matrices. However, realistic systems often exhibit complex, nonlinear associations that linear models may not adequately capture. To overcome these limitations, our framework formalizes nonlinear relationships among variables through node-wise nonlinear regression. Specifically, for the $j$-th variable with group assignment $C_j = l$, consider

$$x_{i,j} = h_j(\boldsymbol{x}_{i,-j}) + \varepsilon_{ij} \triangleq h_j \left( \left( \tilde{\boldsymbol{x}}_{i1}^\top, \ldots, \tilde{\boldsymbol{x}}_{il,-j}^\top, \cdots, \tilde{\boldsymbol{x}}_{iL}^\top \right)^\top \right) + \varepsilon_{ij}, \tag{1}$$

where $\tilde{\boldsymbol{x}}_{il,-j}$ specifically excludes the $j$-th variable from its native group $l$, $\varepsilon_{ij}$ is independent and identically distributed sub-Gaussian stochastic noise with mean zero and sub-Gaussian parameter $\sigma$,

and $h_j : \mathbb{R}^{p-1} \to \mathbb{R}$ is a sparse nonparametric mapping (Figure 1 B). Model (1) captures the nonlinear relationships between $x_{i,j}$ and all other variables $\boldsymbol{x}_{i,-j}$, enabling more accurate characterization of complex system mechanisms than linear approximations permit. The group information is incorporated in (1) through structured group inputs $\tilde{\boldsymbol{x}}_{il}$. This strategy can also accommodate nonlinear high-level relationships between group $l$ and all other groups, leading to deeper investigation of the underlying structures (Figure 1 C).

For estimating $h_j(\cdot)$ and performing bi-level network estimation, we propose a neural network with two hierarchical selection layers and $D$ hidden layers (Figure 1 D):

$$f_{\boldsymbol{\alpha}_j}(\boldsymbol{x}_{i,-j}) = T_j^{(D)} \circ \varrho \circ T_j^{(D-1)} \circ \cdots \circ \varrho \circ T_j^{(1)} \circ \left( \boldsymbol{\Gamma}_j \cdot \left( \boldsymbol{\theta}_l \odot \left( \tilde{\boldsymbol{x}}_{i1}^\top, \ldots, \tilde{\boldsymbol{x}}_{il,-j}^\top, \cdots, \tilde{\boldsymbol{x}}_{iL}^\top \right)^\top \right) \right), \quad (2)$$

with $\boldsymbol{\alpha}_j = \left\{ \boldsymbol{\theta}_l, \boldsymbol{\Gamma}_j, \{\boldsymbol{\Delta}_j^{(d)}, \boldsymbol{b}_j^{(d)}\}_{d=1}^D \right\}$ being the parameter set. Here, each affine transformation is given by $T_j^{(d)}(\boldsymbol{u}) = \boldsymbol{\Delta}_j^{(d)}\boldsymbol{u} + \boldsymbol{b}_j^{(d)}$ with learnable parameters $\boldsymbol{\Delta}_j^{(d)} \in \mathbb{R}^{w_j^{(d)} \times w_j^{(d-1)}}$ and $\boldsymbol{b}_j^{(d)} \in \mathbb{R}^{w_j^{(d)} \times 1}$, where $w_j^{(d)}$ is the width of the $d$-th hidden layer. The activation function $\varrho$ is chosen as the ReLU function and $\odot$ is the block-wise product defined in *Notation*.

In (2), we innovatively introduce two selection layers. Specifically, the first selection layer involves the vector $\boldsymbol{\theta}_l = (\theta_{l1}, \ldots, \theta_{ll}, \ldots, \theta_{lL})^\top$, which encodes group-level (high-level) dependencies with $\theta_{ll} = 1$. Simultaneously, the second selection layer involves the matrix $\boldsymbol{\Gamma}_j \in \mathbb{R}^{w_j^{(1)} \times (p-1)}$, which captures specific associations between variable $j$ and the other $p-1$ variables. Hence, by integrating these hierarchical components, the relationships among low-level variables are explicitly characterized by $\boldsymbol{\theta}_l \odot \boldsymbol{\Gamma}_j^\top$, where high-level group dependencies act as latent scaffolds that constrain or enable low-level dependencies.

Denote $\boldsymbol{\alpha} = \left\{ \boldsymbol{\theta}, \left\{ \boldsymbol{\Gamma}_j, \{\boldsymbol{\Delta}_j^{(d)}, \boldsymbol{b}_j^{(d)}\}_{d=1}^D \right\}_{j=1}^p \right\}$ with $\boldsymbol{\theta} = (\boldsymbol{\theta}_1, \boldsymbol{\theta}_2, \cdots, \boldsymbol{\theta}_L)$, and $\boldsymbol{\Gamma}_j^{[:,k]}$ as the $k$-th column of $\boldsymbol{\Gamma}_j$. Integrating $f_{\boldsymbol{\alpha}_1}(\boldsymbol{x}_{i,-1}), \cdots, f_{\boldsymbol{\alpha}_p}(\boldsymbol{x}_{i,-p})$, we propose a dual-penalized estimator designed to achieve bi-level sparsity for network estimation:

$$\hat{\boldsymbol{\alpha}}_n = \underset{\boldsymbol{\alpha}}{\operatorname{argmin}} \ \sum_{j=1}^p \frac{1}{n} \sum_{i=1}^n l(\boldsymbol{\alpha}_j, x_{i,j}, \boldsymbol{x}_{i,-j}) + \lambda_1 \sum_{l<l'} |\theta_{ll'}| + \lambda_2 \sum_{j=1}^p \sum_{k=1}^{p-1} \left\| \boldsymbol{\Gamma}_j^{[:,k]} \right\|, \quad (3)$$

with $l(\boldsymbol{\alpha}_j, x_{i,j}, \boldsymbol{x}_{i,-j}) = \left( x_{i,j} - f_{\boldsymbol{\alpha}_j}(\boldsymbol{x}_{i,-j}) \right)^2$ and $\| \cdot \|$ being the $L_2$-norm of a vector.

Here, the first term constitutes a quadratic reconstruction loss. The second term implements a Lasso penalty on $\theta_{ll'}$, inducing sparsity through element-wise shrinkage towards zero, which enables automatic high-level edge selection. This foundational assumption of inter-group sparsity aligns with established practices in prior methodological work [5, 29]. This is well illustrated in the context of genomic networks, where pathway-to-pathway connections exhibit natural sparsity since regulatory relationships occur only between specific pathway pairs rather than universally [27]. The third term employs a group Lasso penalty on $\boldsymbol{\Gamma}_j^{[:,k]}$, enforcing simultaneous shrinkage of entire parameter vectors to zero, thereby facilitating low-level edge selection. The tuning parameters $\lambda_1$ and $\lambda_2$ govern edge selection stringency, with larger values yielding sparser network structures.

Based on $\hat{\boldsymbol{\alpha}}_n$, to mitigate regularization bias and enhance selection consistency, we further introduce an adaptive bi-level sparse estimator:

$$\tilde{\boldsymbol{\alpha}}_n = \underset{\boldsymbol{\alpha}}{\operatorname{argmin}} \ \sum_{j=1}^p \frac{1}{n} \sum_{i=1}^n l(\boldsymbol{\alpha}_j, x_{i,j}, \boldsymbol{x}_{i,-j}) + \zeta_1 \sum_{l<l'} \frac{|\theta_{ll'}|}{|\hat{\theta}_{ll'}|^\gamma} + \zeta_2 \sum_{j=1}^p \sum_{k=1}^{p-1} \frac{\left\| \boldsymbol{\Gamma}_j^{[:,k]} \right\|}{\left\| \hat{\boldsymbol{\Gamma}}_j^{[:,k]} \right\|^\gamma}, \quad (4)$$

with $\zeta_1$ and $\zeta_2$ being two tuning parameters for controlling network sparsity and $\gamma$ being a positive constant. Here, the weights $|\hat{\theta}_{ll'}|^\gamma$ and $\left\| \hat{\boldsymbol{\Gamma}}_j^{[:,k]} \right\|^\gamma$ are introduced to adaptively reduce penalties for connections with larger initial estimates while amplifying the shrinkage for smaller ones, resulting in more accurate edge selection owing to reduced estimation bias.

The final bi-level network is constructed based on $\tilde{\boldsymbol{\alpha}}_n$. Specifically, high-level connections between groups $l$ and $l'$ are established when either $\tilde{\theta}_{ll'} \neq 0$ or $\tilde{\theta}_{l'l} \neq 0$. This high-level connectivity then

informs low-level interactions: an edge forms between node $j$ in group $l$ and node $k$ in group $l'$ if either $\tilde{\theta}_{ll'}\tilde{\mathbf{\Gamma}}_j^{[:,k]} \neq \mathbf{0}$ or $\tilde{\theta}_{l'l}\tilde{\mathbf{\Gamma}}_k^{[:,j]} \neq \mathbf{0}$. The proposed bi-level network estimation strategy can also be extended to accommodate overlapping groups, with the details provided in the Appendix.

## 3 Statistical Properties

### 3.1 Approximation Error

Without loss of generality, assume that $\boldsymbol{x}_i \in \mathcal{X} \subset [0,1]^p$. Define the $\beta$-Hölder smooth class:

$$\mathcal{H}^\beta([0,1]^s, B_0) = \left\{ f : [0,1]^s \to \mathbb{R}, \max_{\|\alpha\|_1 \leq \lfloor\beta\rfloor} \|\partial^\alpha f\|_\infty \leq B_0, \max_{\|\alpha\|_1 = \lfloor\beta\rfloor} \sup_{x \neq y} \frac{|\partial^\alpha f(x) - \partial^\alpha f(y)|}{\|x-y\|_2^{\beta - \lfloor\beta\rfloor}} \leq B_0 \right\},$$

where $\partial^\alpha = \partial^{\alpha_1} \cdots \partial^{\alpha_s}$ with $\alpha = (\alpha_1, \ldots, \alpha_s)^\top \in \mathbb{N}_0^s$ and $\mathbb{N}_0$ denotes the set of non-negative integers, $\|\alpha\|_1 = \sum_{i=1}^s |\alpha_i|$ and $\lfloor\beta\rfloor$ denotes the largest integer strictly smaller than $\beta$. If a function belongs to $\mathcal{H}^\beta([0,1]^s, B_0)$, then all the partial derivatives up to order $\lfloor\beta\rfloor$ exist, and the partial derivatives of order $\lfloor\beta\rfloor$ are $\beta - \lfloor\beta\rfloor$ Hölder continuous.

**Asummption 3.1.** *For $j = 1, \cdots, p$, the target sparse function $h_j$ resides in the $s_j$-sparse function class $\mathcal{F}_{s_j}$, where $\mathcal{F}_s = \left\{ h : [0,1]^{p-1} \to [0,1] : \exists \bar{h} : [0,1]^s \to [0,1] \in \mathcal{H}^\beta([0,1]^s, B_0) \right.$ $\left. s.t.\ h(\boldsymbol{x}) = \bar{h}(\tilde{\boldsymbol{x}}), \forall \boldsymbol{x} \in [0,1]^{p-1} \right\}$, with $\tilde{\boldsymbol{x}} \in \mathbb{R}^s$ representing the relevant $s$-dimensional subvector of $\boldsymbol{x}$.*

Assumption 3.1 posits that the target functions exhibit sparsity and smoothness. This is a common requirement in neural network approximation theory [15, 4]. In addition, the assumption is grounded in reality: for instance, in gene regulatory networks, genes are sparsely rather than fully connected, and nonlinear effects vary smoothly rather than abruptly.

Next, we first establish the approximation capabilities of our proposed sparse ReLU feedforward neural network architecture for sparse nonlinear functions. We define the width of a neural network as the maximum width among its hidden layers.

**Theorem 3.2** (Approximation Error). *If $h(\boldsymbol{x}) \in \mathcal{F}_s$, then for any positive integer $N$ and $M$, there exists a sparse ReLU feedforward neural network $f(\boldsymbol{x})$ with width $W = 38(\lfloor\beta\rfloor + 1)^2 3^s s^{\lfloor\beta\rfloor+1} N\lceil\log_2(8N)\rceil$, depth $D = 21(\lfloor\beta\rfloor + 1)^2 M\lceil\log_2(8M)\rceil + 2s$, and the selection layer parameter $(\boldsymbol{\theta} \odot \mathbf{\Gamma}^\top)^\top$ being able to be rearranged as $[\boldsymbol{u}^{w^{(1)} \times s}, \mathbf{0}^{w^{(1)} \times (p-1-s)}]$ such that:*

1. *Support equivalence: both $h(\boldsymbol{x})$ and $f(\boldsymbol{x})$ are $s$-sparse functions and have the same support;*

2. *Uniform approximation:*

$$\sup_{\|\boldsymbol{x}\| \leq [0,1]^{p-1}} |f(\boldsymbol{x}) - h(\boldsymbol{x})| \leq 19 B_0 (\lfloor\beta\rfloor + 1)^2 s^{\lfloor\beta\rfloor + (\beta \vee 1)/2} (NM)^{-2\beta/s}. \tag{5}$$

Under Assumption 3.1, Theorem 3.2 indicates that sparser nonlinear functions require smaller network dimensions for accurate approximation—an observation aligning with intuition. Therefore, the target sparse function $h_j$ can be approximated by a sparse ReLU feedforward network with parameter set $\boldsymbol{\alpha}_j^*$, denoted as $f_{\boldsymbol{\alpha}_j^*}$.

### 3.2 Bi-level Selection Consistency

Let $c_j : \mathbb{N} \to \mathbb{N}$ be the column label mapping where $c_j(k)$ denotes the column of $\mathbf{\Gamma}_j$ corresponding to the $k$-th variable, that is, $c_j(k) = k$ if $k < j$ and $c_j(k) = k - 1$ if $k > j$. Denote $[p]$ and $[L]$ as the set $\{1, \cdots, p\}$ and $\{1, \cdots, L\}$, respectively, and $\mathcal{A}^c$ as the complement of $\mathcal{A}$. In addition, we formalize three fundamental sets: $\mathcal{A}_j = \left\{ k \in [p] : \theta_{C_j, C_k}^* \neq 0 \text{ and } \mathbf{\Gamma}_j^{*[:, c_j(k)]} \neq 0 \right\}$, $\mathcal{B}_j = \left\{ k \in [p] : \mathbf{\Gamma}_j^{*[:, c_j(k)]} = 0 \right\}$, and $\mathcal{P}_l = \left\{ l' \in [L] : \theta_{l,l'}^* = 0 \right\}$.

Then, we can define the bi-level selection consistency as follows.

**Definition 3.3** (Bi-level Selection Consistency). *We say the estimator $(\widetilde{\boldsymbol{\theta}}, \{\widetilde{\mathbf{\Gamma}}_j\}_{j=1}^p)$ achieves bi-level selection consistency if $\forall \delta > 0, \exists N_\delta \in \mathbb{N}$ such that $\forall n > N_\delta$:*

1. **High-level true positives:** $\widetilde{\theta}_{ll'} \neq 0$ for all $(l, l') \in [L] \times \mathcal{P}_l^c$;

2. **High-level true negatives:** $\widetilde{\theta}_{ll'} = 0$ for all $(l, l') \in [L] \times \mathcal{P}_l$;

3. **Low-level true positives:** $\left(\left(\widetilde{\boldsymbol{\theta}}_{C_j} \odot \widetilde{\boldsymbol{\Gamma}}_j^\top\right)^\top\right)^{[:,c_j(k)]} \neq 0$ for all $(j, k) \in [p] \times \mathcal{A}_j$;

4. **Low-level true negatives:** $\left(\left(\widetilde{\boldsymbol{\theta}}_{C_j} \odot \widetilde{\boldsymbol{\Gamma}}_j^\top\right)^\top\right)^{[:,c_j(k)]} = 0$ for all $(j, k) \in [p] \times \mathcal{A}_j^c$;

with probability at least $1 - \delta$.

Let $\boldsymbol{\alpha}^*$ consist of all $\boldsymbol{\alpha}_1^*, \cdots, \boldsymbol{\alpha}_p^*$ and $\mathcal{W}$ denote the feasible parameter space with $\boldsymbol{\alpha}^* \in \mathcal{W}$. We define the population risk $R(\boldsymbol{\alpha})$ and empirical risk $R_n(\boldsymbol{\alpha})$ associated with the squared error as:

$$R(\boldsymbol{\alpha}) = \sum_{j=1}^p R_j(\boldsymbol{\alpha}_j) = \sum_{j=1}^p \mathbb{E}\left[\left(f_{\boldsymbol{\alpha}_j}(\boldsymbol{x}_{-j}) - x_j\right)^2\right], \tag{6}$$

$$R_n(\boldsymbol{\alpha}) = \sum_{j=1}^p R_{nj}(\boldsymbol{\alpha}_j) = \frac{1}{n} \sum_{j=1}^p \sum_{i=1}^n \left(f_{\boldsymbol{\alpha}_j}(\boldsymbol{x}_{i,-j}) - x_{ij}\right)^2. \tag{7}$$

The optimal parameter set is defined as:

$$\mathcal{H}^* = \{\boldsymbol{\alpha} \in \mathcal{W} : R(\boldsymbol{\alpha}) = R(\boldsymbol{\alpha}^*)\}. \tag{8}$$

Despite the intricate geometric structure of $\mathcal{H}^*$, we establish in Lemma A.1 (Appendix) the fundamental equivalence:

$$\boldsymbol{\alpha}_0 \in \mathcal{H}^* \iff f_{\boldsymbol{\alpha}_{0j}} = f_{\boldsymbol{\alpha}_j^*}, \quad \forall j \in \{1, \ldots, p\}. \tag{9}$$

This equivalence implies that both the estimation and selection consistency can be simultaneously attained by controlling the proximity of the parameters to the optimal set $\mathcal{H}^*$. Define $d(\boldsymbol{\alpha}, \mathcal{H}^*) = \inf_{\boldsymbol{\beta} \in \mathcal{H}^*} \|\boldsymbol{\alpha} - \boldsymbol{\beta}\|$.

To establish these theoretical guarantees, we introduce the following assumptions.

**Asummption 3.4.** *Define $\mathcal{F}$ as a class of ReLU feedforward neural networks $f_{\check{\boldsymbol{\alpha}}} : [0, 1]^{p-1} \to [0, 1]$ with parameter $\check{\boldsymbol{\alpha}}$, depth $D$, width $W$, size $S$ (the number of elements in $\check{\boldsymbol{\alpha}}$) and $B$-Lipschitz continuity. We assume that $f_{\boldsymbol{\alpha}_j} \in \mathcal{F}$ for all $j = 1, 2, \ldots, p$.*

**Asummption 3.5.** *There exist $c_2 > 0$ and $\nu > 2$ such that $R(\boldsymbol{\beta}) - R(\boldsymbol{\alpha}^*) \geq c_2 d(\boldsymbol{\beta}, \mathcal{H}^*)^\nu$ for all $\boldsymbol{\beta} \in \mathcal{W}$.*

Assumption 3.4 posits the boundedness of the ReLU neural network function class. This assumption holds in practice, as real-world data (e.g., gene expression levels) and weights are naturally bounded by physical constraints. Assumption 3.5 is a technical assumption. For a fixed network with an analytic activation function, it holds and can be justified by Lojasiewicz's inequality [9].

Under Assumptions 3.4 and 3.5, the following convergence properties hold:

**Theorem 3.6** (Group Lasso + Lasso Convergence)**.** *Let $p = o(\log n)$, $SD \log(S) = O(n^{\frac{1}{4}})$, $\lambda_1 = O(n^{-\frac{1}{8}})$, and $\lambda_2 = O(n^{-\frac{1}{8}})$, then there exist $c_3 > 0$ and $c_4 > 0$ such that*

$$d(\hat{\boldsymbol{\alpha}}_n, \mathcal{H}^*) \leq c_3 \left(\frac{\log n}{n^{\frac{1}{8}}}\right)^{\frac{1}{\nu-1}} \tag{10}$$

*and*

$$\sum_{j=1}^p \sum_{k \in \mathcal{B}_j} \left\|\hat{\boldsymbol{\Gamma}}_j^{[:,c_j(k)]}\right\| + \sum_{l=1}^L \sum_{l' \in \mathcal{P}_l} |\hat{\theta}_{l,l'}| \leq c_4 \log n \left(\frac{\log n}{n^{\frac{1}{8}}}\right)^{\frac{1}{\nu-1}} \tag{11}$$

*holds with probability at least $1 - \delta_1$ with $\delta_1 = 4n\left(n^{\frac{1}{4}} + 1\right)^{\log n} \left(32en^{\frac{1}{4}}\right)^{(\log n)n^{\frac{1}{4}}} e^{-\frac{\sqrt{n}\log n}{32}}$.*

**Theorem 3.7** (Consistency of the adaptive bi-level sparse estimator)**.** *Let $\gamma > 0, \epsilon > 0, \lambda_1 = O(n^{-\frac{1}{8}}), \lambda_2 = O(n^{-\frac{1}{8}}), \zeta_1 = O\left(n^{-\frac{\gamma}{8(\nu-1)}+\epsilon}\right), and \zeta_2 = O\left(n^{-\frac{\gamma}{8(\nu-1)}+\epsilon}\right),$ then the estimator $\tilde{\boldsymbol{\alpha}}_n$ with adaptive bi-level sparse penalty has bi-level selection consistency, and there exists $c_5 > 0$ such that*

$$d(\tilde{\boldsymbol{\alpha}}_n, \mathcal{H}^*) \leq c_5 n^{\left(-\frac{\gamma}{8(\nu-1)}+\epsilon\right)/\nu}, \tag{12}$$

*holds with probability at least $1 - \delta_1$.*

Under Assumption 3.4 and Theorem 3.7, the estimation error can be bounded by leveraging the Lipschitz continuity $|f_{\tilde{\boldsymbol{\alpha}}_j} - f_{\boldsymbol{\alpha}_j^*}| \leq Bc_5 n^{\left(-\frac{\gamma}{8(\nu-1)}+\epsilon\right)/\nu}$. This result implies that the estimation error decays as sample size $n$ increases. Combined Theorem 3.2 and Theorem 3.7, we can easily establish the estimation consistency of $f_{\tilde{\boldsymbol{\alpha}}_j}$ by triangle inequality. That is, $|f_{\tilde{\boldsymbol{\alpha}}_j} - h_j| \leq Bc_5 n^{\left(-\frac{\gamma}{8(\nu-1)}+\epsilon\right)/\nu} + 19B_0(\lfloor\beta\rfloor + 1)^2 s^{\lfloor\beta\rfloor+(\beta\vee 1)/2}(NM)^{-2\beta/s}$ with probability at least $1 - \delta_1$.

## 4 Computation

The proximal gradient descent algorithm for training NNBLNet is outlined in Algorithm 1 (see Appendix). Following the convergence conditions in Theorems 3.6 and 3.7, we set $\lambda_k = \zeta_k = c \cdot n^{-1/8}$ for $k = 1, 2$, with $c$ as a tunable constant. This choice follows conventional rate settings in high-dimensional sparse inference [22]. Empirically, we recommend $c = 0.35$, as it achieves satisfactory accuracy across diverse data settings. For finer calibration, these parameters can be selected via cross-validation (see Appendix A.4). Consistent with common practice, we set $\gamma = 1$. Regarding neural network hyperparameters, exploratory experiments (Appendix A.4) showed that a configuration of 1000 training epochs, three hidden layers, and 50 nodes per layer offers an optimal balance between accuracy and computational efficiency. This fixed setup was used in all experiments to ensure reproducibility and consistency, in line with common practice [33].

## 5 Experiment

To comprehensively validate our methodology, we established a dual evaluation framework encompassing both synthetic benchmarks and real-world networks. The synthetic analysis included systematically constructed networks with predefined nonlinear and linear patterns, enabling a controlled assessment of relationship modeling capabilities. For real-world validation, we analyzed four distinct network types: (1) Friendship– social connections among high school students from a high school; (2) Co-authorship–collaborative relationships in academic publications; (3-4) BRCA and LUAD–gene regulatory networks derived from The Cancer Genome Atlas (TCGA) breast cancer and lung adenocarcinoma data, respectively.

In addition to the proposed NNBLNet, we conducted systematic comparisons with three alternative methodologies: (1) BGSL [7], a Bayesian Gaussian graphical modeling framework with explicit group-structured variable representations; (2) Fair Glasso [25], which enhances sparse precision matrix estimation through group-aware regularization constraints; and (3) DeepGRNCS [19], a multi-task deep learning architecture specializing in joint network inference. Experimental details, including the synthetic settings and introductions to the real-world datasets, are provided in Appendix A.5. To supplement the results presented in the main text, we also conducted a series of sensitivity analyses in Appendix A.6, including an assessment of model generalization capability based on the stability of network estimation and performance comparisons under different sample sizes, group sizes, and group label misclassification rates, as well as downstream analysis of the LUAD dataset.

### 5.1 Synthetic Networks with Nonlinear and Linear signals

In Table 1, NNBLNet consistently outperformed all baseline methods in nonlinear scenario, achieving the highest F1-score of 0.772. This improvement was attributed to its hierarchical design, which jointly modeled both within-group and between-group edges through bi-level regularization. Fair Glasso and BGSL, which impose sparsity within the GGM framework, yielded comparable results with F1-scores of 0.710 and 0.709, respectively. DeepGRNCS, a deep learning-based method, performed better than classical approaches but fell short of our proposed method, suggesting that

integrating bi-level selection is crucial for recovering multiscale network structure. These results demonstrate the efficacy of NNBLNet in uncovering meaningful networks with greater accuracy and structural coherence.

Under the linear setting, Fair Glasso attained the highest F1-score (0.790), aligning with its design premise for sparse Gaussian graphical models. NNBLNet remained highly competitive (F1 = 0.769), which illustrates its adaptability across both linear and nonlinear data regimes. Although specifically designed for hierarchical nonlinear dependencies, it maintained robust performance even in simpler settings. DeepGRNCS also delivered favorable results, while BGSL achieved a balanced recall-precision trade-off. These findings verify that NNBLNet not only performs stably in conventional linear estimation tasks but also offers flexibility to more complex environments.

Table 1: Performance comparison of different methods for the synthetic network with nonlinear and linear relationships: Mean (SD) over 100 replicates.

| Patterns | Metric | NNBLNet | Fair Glasso | BGSL | DeepGRNCS |
|----------|--------|---------|-------------|------|-----------|
| Nonlinear | Recall | 0.872(0.016) | 0.779(0.014) | 0.790(0.017) | 0.809(0.016) |
| | Precision | 0.693(0.022) | 0.653(0.021) | 0.644(0.018) | 0.681(0.018) |
| | F1-score | 0.772(0.014) | 0.710(0.017) | 0.709(0.013) | 0.731(0.017) |
| Linear | Recall | 0.881(0.018) | 0.901(0.016) | 0.836(0.016) | 0.846(0.016) |
| | Precision | 0.675(0.020) | 0.704(0.019) | 0.692(0.019) | 0.671(0.020) |
| | F1-score | 0.765(0.013) | 0.790(0.013) | 0.757(0.015) | 0.763(0.017) |

## 5.2 Four Real-World Networks

We further evaluated our method on four real-world datasets spanning social networks and biological systems: Friendship, Co-authorship, BRCA, and LUAD. Performance results are summarized in Table 2.

Table 2: Performance comparison of different methods for the four real-world networks.

| Dataset | Metric | NNBLNet | Fair Glasso | BGSL | DeepGRNCS |
|---------|--------|---------|-------------|------|-----------|
| Friendship | Recall | 0.875 | 0.859 | 0.798 | 0.811 |
| | Precision | 0.804 | 0.703 | 0.735 | 0.686 |
| | F1 | 0.838 | 0.771 | 0.765 | 0.745 |
| Co-authorship | Recall | 0.712 | 0.643 | 0.622 | 0.655 |
| | Precision | 0.678 | 0.606 | 0.591 | 0.610 |
| | F1 | 0.695 | 0.619 | 0.604 | 0.632 |
| BRCA | Recall | 0.764 | 0.812 | 0.676 | 0.742 |
| | Precision | 0.618 | 0.652 | 0.552 | 0.601 |
| | F1 | 0.683 | 0.723 | 0.607 | 0.664 |
| LUAD | Recall | 0.641 | 0.625 | 0.597 | 0.654 |
| | Precision | 0.526 | 0.517 | 0.489 | 0.539 |
| | F1 | 0.577 | 0.566 | 0.537 | 0.591 |

Across all four datasets, NNBLNet consistently achieved either the best or second-best F1-score, demonstrating its versatility and robustness in diverse application domains. On the Friendship network, it significantly outperformed all competitors with an F1-score of 0.838, reflecting its strength in modeling community-driven structures. In the Co-authorship setting, where overlapping communities and latent hierarchies were expected, NNBLNet again led with an F1-score of 0.695, significantly outperforming Fair Glasso (0.619), BGSL (0.604), and DeepGRNCS (0.632). These results indicate the advantage of jointly modeling inter-group and intra-group dependencies.

For the biological datasets, BRCA and LUAD, NNBLNet maintained top-tier performance. While Fair Glasso achieved a slightly higher F1 score in BRCA (0.723), NNBLNet offered more balanced precision-recall trade-offs, especially in LUAD, where it achieved the second highest F1-score

(0.577). The improvement in biological contexts suggests that our method effectively captures complex relationships between molecular factors, including both nonlinear individual factor-level dynamics and pathway-level crosstalk.

Collectively, these results demonstrate that NNBLNet is a highly competitive and generalizable framework for network estimation, performing consistently well across domains with varying complexity and noise characteristics.

### 5.3 Ablation Study

To validate the necessity of the proposed bi-level architecture, we performed ablation studies comparing our model against two simplified variants: the flat-structured NNNet, which ignores the bi-level design, and Modified-NNNet, which uses group labels but omits intra-group adjacency modeling. As shown in Table 3, NNBLNet consistently outperformed both variants on synthetic and real-world datasets. The significant performance gain confirms that the bi-level structure is essential for achieving higher recall, precision, and F1 scores.

To further isolate the benefits of the bi-level architecture beyond group information alone, we conducted sensitivity analyses in scenarios with extreme sparsity of inter-group edges (Table 4), where the ratio of inter-group edges to total edges ($\eta$) was set to 0%, 0.5%, and 1%. The results showed that NNBLNet consistently achieved the highest F1 by balancing recall and precision. In addition, the modified-NNNet failed to capture inter-group edges and exhibited declining recall as $\eta$ increased, while the NNNet, which ignored inter-group sparsity, produced excessive false positives under sparse settings. These findings confirmed that the bi-level hierarchical sparsity structure is essential for accurate network inference.

Table 3: Performance comparison of NNBLNet, NNNet, and Modified-NNNet across six datasets (mean values and standard deviation for synthetic datasets).

| Dataset | Method | Recall | Precision | F1 |
|---|---|---|---|---|
| Nonlinear | NNBLNet | 0.872 (0.016) | 0.693 (0.022) | 0.772 (0.014) |
| | NNNet | 0.844 (0.017) | 0.656 (0.019) | 0.738 (0.014) |
| | Modified-NNNet | 0.782 (0.020) | 0.660 (0.021) | 0.716 (0.017) |
| Linear | NNBLNet | 0.881 (0.018) | 0.675 (0.020) | 0.769 (0.013) |
| | NNNet | 0.857 (0.014) | 0.628 (0.017) | 0.727 (0.013) |
| | Modified-NNNet | 0.802 (0.018) | 0.627 (0.019) | 0.704 (0.016) |
| Friendship | NNBLNet | 0.875 | 0.804 | 0.838 |
| | NNNet | 0.842 | 0.745 | 0.790 |
| | Modified-NNNet | 0.774 | 0.734 | 0.753 |
| Co-authorship | NNBLNet | 0.712 | 0.678 | 0.695 |
| | NNNet | 0.674 | 0.637 | 0.649 |
| | Modified-NNNet | 0.612 | 0.636 | 0.624 |
| BRCA | NNBLNet | 0.764 | 0.618 | 0.683 |
| | NNNet | 0.709 | 0.563 | 0.627 |
| | Modified-NNNet | 0.655 | 0.566 | 0.607 |
| LUAD | NNBLNet | 0.641 | 0.526 | 0.577 |
| | NNNet | 0.613 | 0.502 | 0.552 |
| | Modified-NNNet | 0.552 | 0.503 | 0.527 |

## 6 Discussion

This work presents NNBLNet, a neural network framework for bi-level network inference. The method is built upon a key structural prior that represents dependency structures in complex systems as hierarchical. This hierarchical modeling is implemented through two mechanisms: intra-group information sharing, which amplifies weak signals via latent pooling within groups, and inter-group sparse transmission, which gates cross-group connections through switches ($\theta_{ll'}$) to suppress irrelevant noise. NNBLNet represents a paradigm shift in structured network inference by unifying neural network representation learning with hierarchical sparsity constraints. Its innovative hierarchical

Table 4: Performance of NNBLNet, NNNet, and Modified-NNNet under sparse inter-group edges (mean and standard deviation over 100 replicates).

| Setting | $\eta$ | Method | Recall | Precision | F1 |
|---|---|---|---|---|---|
| Nonlinear | 0% | NNBLNet | 0.752 (0.018) | 0.684 (0.020) | 0.716 (0.017) |
| | | NNNet | 0.744 (0.019) | 0.619 (0.023) | 0.676 (0.018) |
| | | Modified-NNNet | 0.737 (0.018) | 0.682 (0.020) | 0.708 (0.016) |
| | 0.5% | NNBLNet | 0.760 (0.017) | 0.696 (0.019) | 0.727 (0.016) |
| | | NNNet | 0.752 (0.018) | 0.616 (0.022) | 0.677 (0.017) |
| | | Modified-NNNet | 0.717 (0.019) | 0.673 (0.021) | 0.694 (0.017) |
| | 1% | NNBLNet | 0.774 (0.016) | 0.715 (0.018) | 0.743 (0.014) |
| | | NNNet | 0.764 (0.018) | 0.615 (0.023) | 0.681 (0.019) |
| | | Modified-NNNet | 0.696 (0.020) | 0.663 (0.022) | 0.679 (0.019) |
| Linear | 0% | NNBLNet | 0.740 (0.019) | 0.672 (0.021) | 0.704 (0.018) |
| | | NNNet | 0.732 (0.020) | 0.607 (0.024) | 0.664 (0.019) |
| | | Modified-NNNet | 0.724 (0.019) | 0.671 (0.021) | 0.696 (0.017) |
| | 0.5% | NNBLNet | 0.750 (0.018) | 0.684 (0.020) | 0.715 (0.017) |
| | | NNNet | 0.741 (0.019) | 0.605 (0.023) | 0.666 (0.018) |
| | | Modified-NNNet | 0.706 (0.020) | 0.662 (0.021) | 0.682 (0.018) |
| | 1% | NNBLNet | 0.766 (0.017) | 0.702 (0.019) | 0.733 (0.015) |
| | | NNNet | 0.753 (0.019) | 0.605 (0.024) | 0.671 (0.019) |
| | | Modified-NNNet | 0.681 (0.021) | 0.652 (0.022) | 0.666 (0.020) |

selection layer explicitly captures bi-level dependencies: local associations between individual variables and global coordination among groups. This architecture effectively models complex nonlinear relationships while maintaining interpretability. Theoretically, we establish a bridge between neural modeling and statistical guarantees by proving estimation consistency and exact bi-level selection consistency. Empirical evaluations and real-data analyses demonstrate NNBLNet's effectiveness, showing it achieves superior F1 scores compared to competing methods.

## 6.1 Limitation

This study has several limitations that point to valuable directions for future research. Theoretically, our analysis relies on the assumption of sub-Gaussian noise, which, while common in statistical literature, is often difficult to verify in practical applications. Despite this limitation, empirical comparisons against authoritative ground-truth networks demonstrate that our method still achieves competitive network reconstruction accuracy, thereby offering partial validation of its practical effectiveness. Moreover, our method relies on pre-defined group labels, and performance may decline if these are noisy or incomplete. Future extensions could jointly infer group memberships and associations or incorporate unsupervised techniques (e.g., spectral clustering) when prior labels are unavailable, enhancing robustness and applicability across diverse scenarios.

## 6.2 Broader Impact

NNBLNet bridges neural network learning and hierarchical sparse inference to enable interpretable bi-level network discovery across scientific domains, from identifying gene-pathway interplay in disease mechanisms to modeling individual-group dynamics in social systems. Its adaptive sparsity design balances predictive power with mechanistic interpretability, offering actionable insights for precision medicine and policy-making.

## Acknowledgments and Disclosure of Funding

We thank the Area Chair and the anonymous reviewers for their insightful feedback, which was instrumental in strengthening this paper. This research was supported by the National Natural Science Foundation of China (12071273); MOE Project of Humanities and Social Sciences (25YJCZH291); Shanghai Rising-Star Program (22QA1403500); Shanghai Science and Technology Development

Funds (23JC1402100); Shanghai Research Center for Data Science and Decision Technology; National Institutes of Health (CA204120); and National Science Foundation (2209685).

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

# A  Appendix / Supplemental Material

## A.1  Related Works

In recent years, the explosive growth of high-dimensional data, such as high-throughput omics data, has generated unprecedented volumes of complex data, thereby elevating the importance of network estimation research. Among the various existing approaches, Gaussian Graphical Models (GGMs) have emerged as one of the most widely used techniques. In the GGM framework, network structures are inferred by estimating a sparse precision matrix (i.e., the inverse of the covariance matrix), which encodes conditional dependencies among variables conditional on all others [41, 11]. This estimation procedure can be formulated as a series of sparse node-wise linear regressions. Compared to marginal or unconditional correlation-based methods (e.g., Pearson correlation), this conditional strategy offers a more holistic view of system-level dependencies, potentially leading to more meaningful interpretations.

In real-world scenarios, networks often exhibit a bi-level hierarchical structure. This structure implies that some variables are organized into higher-level groups, with lower-level variables nested within these groups. For example, a gene pathway consists of multiple genes that collaborate to perform a specific cellular or physical function. In this context, pathways represent the higher-level groups, while the individual genes within those pathways are the lower-level variables. It is also important to note that these groups are not independent of each other. To tackle hierarchical network estimation, several GGM-based methods have been developed. Cheng et al. (2017) [5] introduced a multilevel Gaussian graphical model for nested data structures. Shan et al. (2020) [29] proposed a framework for joint estimation of two-level GGMs across multiple classes. Colombi et al. (2024) [7] focused on learning block-structured graphical models using variable groupings. Notably, Fair Glasso [25] specifically leverages group information to estimate graphical models with provably unbiased statistical behavior, addressing fairness concerns in network inference. However, these GGM based methods can only capture linear dependencies and may ignore complex nonlinear relationship in the real world.

Deep neural networks (DNNs) and related machine learning models have gained widespread attention due to their strong capacity for nonlinear approximation and representation learning, particularly in high-dimensional settings [21]. These models excel at uncovering complex associations within large datasets, making them particularly appealing for network inference tasks. With the increasing availability of large-scale high-dimensional data, deep learning has become a cornerstone for network estimation. Researchers have recently developed several deep learning-based methods that aim to reconstruct latent networks. These methods harness the expressive power of neural networks to model intricate dependencies between variables, often using architectures such as pre-trained deep neural networks [19], Variational Autoencoders (VAEs) [28] and Graph Convolutional Networks (GCNs) [23]. While promising, these approaches face several important limitations. First, most are supervised and require labeled data—such as group-specific regulatory annotations or curated databases—which are expensive and time-consuming to acquire. Consequently, the size and diversity of training datasets remain limited. Moreover, deep learning methods often lack theoretical guarantees and interpretability, which hampers their adoption in sensitive scientific domains.

To improve interpretability, the research community has proposed a variety of strategies aimed at making deep learning models more transparent and reliable. Among these, regularization-based approaches have shown particular promise. By incorporating additional constraints (e.g., L1 or L2 penalties), regularization can limit model complexity, promote sparsity, and enhance feature selection [30, 20]. This is especially valuable for network inference, where identifying the most influential variables (e.g., hub nodes) is often a primary goal. In scientific applications, regularization enhances both statistical reliability and practical interpretability. Recent theoretical work further supports the effectiveness of regularization for consistent variable selection in deep models [34, 40], providing a rigorous foundation for interpretable deep learning-based network estimation. These theoretical insights not only deepen our understanding of model behavior but also inform the development of robust and generalizable algorithms.

## A.2  Extension to Overlapping Groups

In many scientific applications, variables may simultaneously belong to multiple groups, resulting in overlapping group structures. For example, in genomics, the same gene may participate in several

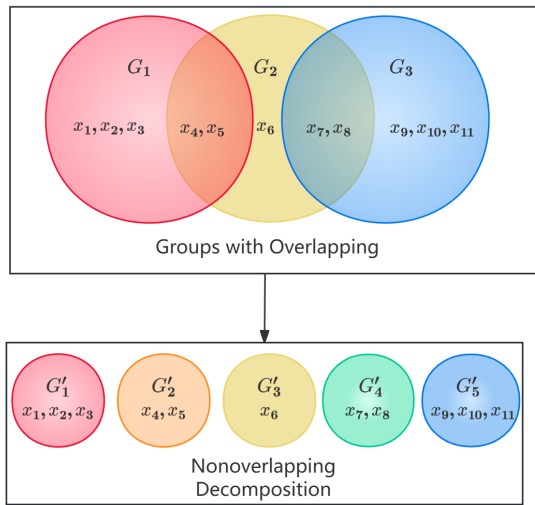

Figure 2: Illustration of overlapping and disjoint group structures. The upper panel shows three pathways $G_1, G_2, G_3$ with shared genes, leading to overlapping groups. The lower panel demonstrates the decomposition of these overlapping pathways into non-overlapping subgroups ($G'_1, \ldots, G'_5$), which allows group-level dependencies to be defined in a consistent and interpretable manner.

biological pathways, so different pathways can share common genes. Such overlaps complicate the definition of group-level dependencies, as conventional methods typically assume disjoint group memberships.

To illustrate, consider three pathways denoted by $G_1$, $G_2$, and $G_3$ as shown in Figure 2. Each contains five genes: $G_1 = \{1,2,3,4,5\}, G_2 = \{4,5,6,7,8\}, G_3 = \{7,8,9,10,11\}$. Here, $G_1$ and $G_2$ overlap on genes $\{4,5\}$, $G_2$ and $G_3$ overlap on $\{7,8\}$, while $G_1$ and $G_3$ have no common elements. To resolve overlaps, we decompose the original groups into disjoint subgroups as shown in Figure 2: $G'_1 = \{1,2,3\}, G'_2 = \{4,5\}, G'_3 = \{6\}, G'_4 = \{7,8\}, G'_5 = \{9,10,11\}$. Accordingly, the original pathways can be represented as $G_1 = \{G'_1, G'_2\}, G_2 = \{G'_2, G'_3, G'_4\}, G_3 = \{G'_4, G'_5\}$. We denote $\theta'_{ll'} = 0$ whenever two disjoint subgroups $G'_l$ and $G'_{l'}$ are conditionally independent. Based on this decomposition, we define the conditional dependency rules among the original pathways:

- Non-overlapping groups: For instance, $G_1$ and $G_3$ have no overlap. If $\theta'_{14} = \theta'_{15} = \theta'_{24} = \theta'_{25} = 0$, then $G_1$ and $G_3$ are conditionally independent, as all variables in $G_1$ are independent of those in $G_3$. Otherwise, they are conditionally dependent.

- Overlapping groups: For adjacent groups such as $G_1$ and $G_2$, if $\theta'_{13} \neq 0$ or $\theta'_{14} \neq 0$, they are conditionally dependent, since at least one unique variable in $G_1$ depends on a unique variable in $G_2$. If instead $\theta'_{13} = \theta'_{14} = 0$, but at least one of $\theta'_{12}, \theta'_{23}, \theta'_{24}$ is nonzero, dependency arises through the common subgroup. Otherwise, $G_1$ and $G_2$ are conditionally independent.

This decomposition preserves interpretability by mapping subgroup-level dependencies back to the original overlapping groups, while avoiding inflated false discoveries due to redundant memberships.

### A.3 Proof of Statistical Properties

### A.3.1 Proof of Theorem 3.2

Since $h(\boldsymbol{x}) \in \mathcal{F}_s$, there exists a $\bar{h}(\tilde{\boldsymbol{x}}) \in \mathcal{H}^\beta([0,1]^s, B_0)$ such that $h(\boldsymbol{x}) = \bar{h}(\tilde{\boldsymbol{x}})$. Refer to Corollary 3.1 in [15], there exists a function $\bar{f}(\tilde{\boldsymbol{x}})$ implemented by a ReLU network with width $W = 38(\lfloor\beta\rfloor + 1)^2 3^s s^{\lfloor\beta\rfloor+1} N\lceil\log_2(8N)\rceil$ and depth $D = 21(\lfloor\beta\rfloor + 1)^2 M\lceil\log_2(8M)\rceil + 2s$ such that

$$|\bar{h}(\tilde{\boldsymbol{x}}) - \bar{f}(\tilde{\boldsymbol{x}})| \leq 19B_0(\lfloor\beta\rfloor + 1)^2 s^{\lfloor\beta\rfloor+(\beta\vee1)/2}(NM)^{-2\beta/s}, \tilde{\boldsymbol{x}} \in [0,1]^s. \tag{13}$$

Knowing that $\left(\boldsymbol{\theta} \odot \boldsymbol{\Gamma}^{\top}\right)^{\top}$ can be rearranged as $\left(\boldsymbol{u}^{w^{(1)} \times s}, \boldsymbol{0}^{w^{(1)} \times (p-1-s)}\right)$, we have $\left(\boldsymbol{\theta} \odot \boldsymbol{\Gamma}^{\top}\right)^{\top} \boldsymbol{x} = \boldsymbol{u}\tilde{\boldsymbol{x}}$. Thus, for any low-dimensional Relu network $\bar{f}(\tilde{\boldsymbol{x}})$ there always exists a high-dimensional sparse ReLU network $f(\boldsymbol{x})$ satisfying $\bar{f}(\tilde{\boldsymbol{x}}) = f(\boldsymbol{x})$. This completes the proof.

### A.3.2 Supporting Lemmas

**Lemma A.1.**   *1. There exists $c_0 > 0$ such that $\theta_{\boldsymbol{\alpha} C_j, C_k} \geq c_0$ and $\left\|\boldsymbol{\Gamma}_{\boldsymbol{\alpha}_j}^{[:,c_j(k)]}\right\| \geq c_0$ for all $k \in \mathcal{A}_j$ and $\boldsymbol{\alpha} \in \mathcal{H}^*$.*

*2. Denote $\phi(\boldsymbol{\alpha})$ the vector obtained from $\boldsymbol{\alpha}$ by setting $\boldsymbol{\Gamma}_{\boldsymbol{\alpha}_j}^{[:,c_j(k)]} = 0$ for all $(j,k) \in \bigcup_{j=1}^{p}\{j\} \times \mathcal{B}_j$ and $\theta_{\boldsymbol{\alpha} l,l'} = 0$ for all $(l,l') \in \bigcup_{l=1}^{L}\{l\} \times \mathcal{P}_l$. For $\boldsymbol{\alpha} \in \mathcal{H}^*$, $\phi(\boldsymbol{\alpha})$ also belongs to $\mathcal{H}^*$.*

*Proof.* By Theorem 3.2, we establish the uniform convergence:

$$\left| h_j(\boldsymbol{x}_{-j}) - f_{\boldsymbol{\alpha}_j^*}(\boldsymbol{x}_{-j}) \right| \to 0 \quad \text{as } W, D \to \infty. \tag{14}$$

This implies that $f_{\boldsymbol{\alpha}_j^*}(\boldsymbol{x}_{-j}) \to \mathbb{E}[x_j | \boldsymbol{x}_{-j}]$ serves as the unique minimizer of $R_j(\boldsymbol{\alpha}_j)$. Consequently, the composite risk functional $R(\boldsymbol{\alpha}^*) = \sum_{j=1}^{p} R_j(\boldsymbol{\alpha}_j^*)$ attains its global minimum through coordinate-wise optimization. For any competing parameter $\boldsymbol{\alpha}_0 \in \mathcal{H}^*$, this construction ensures:

$$R_j(\boldsymbol{\alpha}_{0j}) = R_j(\boldsymbol{\alpha}_j^*), \quad \forall j = 1, \ldots, p. \tag{15}$$

The fundamental inequality

$$R_j(\boldsymbol{\alpha}_j^*) = \min_{g} \mathbb{E}\left[(x_j - g(\boldsymbol{x}_{-j}))^2\right] \leq \min_{\alpha_j \in \mathcal{W}_j} R_j(\boldsymbol{\alpha}_j) = R_j(\boldsymbol{\alpha}_{0j}), \tag{16}$$

holds with equality if and only if $f_{\boldsymbol{\alpha}_j} = f_{\boldsymbol{\alpha}_j^*}$ almost surely, where $g$ can be any measurable function from the input space to the real numbers. Therefore, the identifiability condition

$$\boldsymbol{\alpha}_0 \in \mathcal{H}^* \iff f_{\boldsymbol{\alpha}_{0j}} = f_{\boldsymbol{\alpha}_j^*}, \quad \forall j = 1, \ldots, p, \tag{17}$$

follows necessarily.

1: Assuming that no such $c_0$ exists, there exist $\boldsymbol{\alpha}_0 \in \mathcal{H}^*$ and $k \in \mathcal{A}_j$ such that $\left(\boldsymbol{\theta}_{\boldsymbol{\alpha}_0 C_j} \odot \boldsymbol{\Gamma}_{\boldsymbol{\alpha}_0 j}^{\top}\right)^{\top[:,c_j(k)]} = 0$. This means $f_{\boldsymbol{\alpha}_{0j}} = f_{\boldsymbol{\alpha}_j^*}$ does not depend on the related variable $x_k$, which is a contradiction.

2: Denote $\phi(\boldsymbol{\alpha}_j)$ the sub-vector of $\phi(\boldsymbol{\alpha})$ corresponding to $\boldsymbol{\alpha}_j$. Since $\boldsymbol{\alpha} \in \mathcal{H}^*$, we have $f_{\boldsymbol{\alpha}_j^*}(\boldsymbol{x}_{\mathcal{B}_j}, \boldsymbol{x}_{\mathcal{B}_j^c}) = f_{\boldsymbol{\alpha}_j}(\boldsymbol{x}_{\mathcal{B}_j}, \boldsymbol{x}_{\mathcal{B}_j^c}) = f_{\boldsymbol{\alpha}_j}(\boldsymbol{0}, \boldsymbol{x}_{\mathcal{B}_j^c}) = f_{\phi(\boldsymbol{\alpha}_j)}(\boldsymbol{x}_{\mathcal{B}_j}, \boldsymbol{x}_{\mathcal{B}_j^c})$, which implies $\phi(\boldsymbol{\alpha}) \in \mathcal{H}^*$. $\square$

**Lemma A.2.** *Let $p = o(\log n)$, $SD \log S = O(n^{\frac{1}{4}})$, then there exists $c_1 > 0$ such that*

$$|R_n(\boldsymbol{\alpha}) - R(\boldsymbol{\alpha})| \leq c_1 \frac{\log n}{n^{\frac{1}{4}}}, \quad \forall \boldsymbol{\alpha} \in \mathcal{W}, \tag{18}$$

*holds with probability at least $1 - \delta_1$ with $\delta_1 = 4n\left(n^{\frac{1}{4}} + 1\right)^{\log n}\left(32en^{\frac{1}{4}}\right)^{(\log n)n^{\frac{1}{4}}} e^{-\frac{\sqrt{n}\log n}{32}}$.*

*Proof.* Let $l_{f_j}(\boldsymbol{x}_i) = \left(x_{ij} - f_{\boldsymbol{\alpha}_j}(\boldsymbol{x}_{i,-j})\right)^2$ and $l_f(\boldsymbol{x}_i) = \sum_{j=1}^{p} l_{f_j}(\boldsymbol{x}_i)$. For $x \in \mathcal{X}^n$, denote $R_n(f_x) = \frac{1}{n}\sum_{i=1}^{n} l_f(\boldsymbol{x}_i)$ and $R(f) = \mathbb{E}[l_f(\boldsymbol{x})]$.

Define

$$Q = \left\{\boldsymbol{x} \in \mathcal{X}^n : \exists f_{\boldsymbol{\alpha}_1} \cdots f_{\boldsymbol{\alpha}_p} \in \mathcal{F} \text{ s.t. } |R(f) - R_n(f_x)| \geq \varepsilon\right\}, \tag{19}$$

and

$$R = \left\{(\boldsymbol{r}, \boldsymbol{s}) \in \mathcal{X}^n \times \mathcal{X}^n : \exists f_{\boldsymbol{\alpha}_1} \cdots f_{\boldsymbol{\alpha}_p} \in \mathcal{F} \text{ s.t. } |R_n(f_r) - R_n(f_s)| \geq \frac{\varepsilon}{2}\right\}. \tag{20}$$

Since $\{|R(f) - R_n(f_r)| \geq \varepsilon$ and $|R(f) - R_n(f_s)| < \frac{\varepsilon}{2}\} \subset \{|R_n(f_r) - R_n(f_s)| \geq \frac{\varepsilon}{2}\}$, we have

$$
\begin{aligned}
\mathbb{P}(R) &\geq \mathbb{P}\left\{\exists f_{\boldsymbol{\alpha}_1} \cdots f_{\boldsymbol{\alpha}_p} \in \mathcal{F} \text{ s.t. } |R(f) - R_n(f_r)| \geq \varepsilon \text{ and } |R(f) - R_n(f_s)| < \frac{\varepsilon}{2}\right\} \\
&= \int_Q \mathbb{P}\left\{s : \exists f_{\boldsymbol{\alpha}_1} \cdots f_{\boldsymbol{\alpha}_p} \in \mathcal{F}, |R(f) - R_n(f_r)| \geq \varepsilon \text{ and} |R(f) - R_n(f_s)| < \varepsilon/2\right\} d\mathbb{P}(r).
\end{aligned}
\tag{21}
$$

Noting that $f_{\boldsymbol{\alpha}_j}$ maps into $[0,1]$ and $x_{ij} \in [0,1]$, we have $l_{f_j}(\boldsymbol{x}_i) \in [0,1]$ and $l_f(\boldsymbol{x}_i) \in [0,p]$. Hoeffding's inequality for bounded random variables shows that

$$
\mathbb{P}\left(|R(f) - R_n(f_s)| \leq \frac{\varepsilon}{2}\right) \geq 1 - \exp\left(-\frac{n\varepsilon^2}{4p}\right) \geq \frac{1}{2},
\tag{22}
$$

for $n > \frac{4p}{\varepsilon^2}$. Thus, $\mathbb{P}(Q) \leq 2\mathbb{P}(R)$ for $n > \frac{4p}{\varepsilon^2}$.

In order to bound $\mathbb{P}(R)$, the technique of permutation and reduction to a finite class is useful. Denote $T_m$ the set of permutations on $\{1, 2, \ldots, 2n\}$ that switch elements $i$ and $n + i$, for $i$ in some subset of $\{1, 2, \ldots, 2n\}$. A permutation $\sigma$ is chosen uniformly at random from $T_m$. By Lemma 4.5 of [1],

$$
\mathbb{P}(R) = \mathbb{E}\mathbb{P}(\sigma\boldsymbol{x} \in R) \leq \max_{\boldsymbol{x} \in \mathcal{X}^{2n}} \mathbb{P}(\sigma\boldsymbol{x} \in R).
\tag{23}
$$

where the expectation is over $\boldsymbol{x}$ and the probability is over permutations $\sigma$.

Denote $\mathcal{L}_{\mathcal{F}} = \left\{l_f(\boldsymbol{x}) = \sum_{j=1}^p \left(x_j - f_{\boldsymbol{\alpha}_j}(\boldsymbol{x}_{-j})\right)^2 : f_{\boldsymbol{\alpha}_j} \in \mathcal{F}, j = 1, \ldots, p\right\}$. Let $N\left(\varepsilon, \mathcal{F}, L^1(P_n)\right)$ be the covering number of $\mathcal{F}$ under the empirical $L^1(P_n)$ metric with radius $\varepsilon$. For a given sequence $x = (x_1, \ldots, x_n) \in \mathcal{X}^n$, let $\mathcal{F}|_x = \{(f(x_1), \ldots, f(x_n)) : f \in \mathcal{F}\}$ be the subset of $\mathbb{R}^n$. Define the uniform covering number

$$
\mathcal{N}_n\left(\varepsilon, \mathcal{F}, L^1(P_n)\right) = \max\left\{\mathcal{N}\left(\varepsilon, \mathcal{F}|_x, L^1(P_n)\right) : x \in \mathcal{X}^n\right\}.
\tag{24}
$$

Suppose that $\boldsymbol{x} \in \mathcal{X}^{2n}$ and let $\mathcal{T}$ be a minimal $\frac{\varepsilon}{8}$-cover for $\mathcal{L}_{\mathcal{F}}|_x$ with respect to the $L^1(P_n)$ metric. Pick $\mathcal{G} \subset \mathcal{F}$ such that $\mathcal{T} = \mathcal{L}_{\mathcal{G}}|_x$ and $|\mathcal{G}| = |\mathcal{T}|$. Let $\boldsymbol{r} \in \mathcal{X}^n$ and $\boldsymbol{s} \in \mathcal{X}^n$ so that $\boldsymbol{x} = (\boldsymbol{r}, \boldsymbol{s})$. Suppose that $f \in \mathcal{F}$ satisfies $|R_n(f_r) - R_n(f_s)| \geq \frac{\varepsilon}{2}$ and there exists $g \in \mathcal{G}$ such that $\frac{1}{2n} \sum_{i=1}^{2n} |l_f(\boldsymbol{x}_i) - l_g(\boldsymbol{x}_i)| < \frac{\varepsilon}{8}$. Then,

$$
\begin{aligned}
|R_n(g_r) - R_n(g_s)| &= \left|\frac{1}{n}\sum_{i=1}^n l_g(\boldsymbol{x}_i) - \frac{1}{n}\sum_{i=n+1}^{2n} l_g(\boldsymbol{x}_i)\right| \\
&= \left|\frac{1}{n}\sum_{i=1}^n (l_g(\boldsymbol{x}_i) - l_f(\boldsymbol{x}_i)) - \frac{1}{n}\sum_{i=n+1}^{2n} (l_g(\boldsymbol{x}_i) - l_f(\boldsymbol{x}_i)) + R_n(f_r) - R_n(f_s)\right| \\
&\geq |R_n(f_r) - R_n(f_s)| - \\
&\quad \left|\frac{1}{n}\sum_{i=1}^n (l_g(\boldsymbol{x}_i) - l_f(\boldsymbol{x}_i)) - \frac{1}{n}\sum_{i=n+1}^{2n} (l_g(\boldsymbol{x}_i) - l_f(\boldsymbol{x}_i))\right| \\
&\geq |R_n(f_r) - R_n(f_s)| - \frac{1}{n}\sum_{i=1}^{2n} |l_g(\boldsymbol{x}_i) - l_f(\boldsymbol{x}_i)| \\
&> \varepsilon/4.
\end{aligned}
\tag{25}
$$

Thus,

$$
\mathbb{P}(\sigma\boldsymbol{x} \in R) \leq \mathbb{P}\left(\exists g \in \mathcal{G} : \left|\frac{1}{n}\sum_{i=1}^{n}\left(l_g(\boldsymbol{x}_{\sigma(i)}) - l_g(\boldsymbol{x}_{\sigma(n+i)})\right)\right| \geq \varepsilon/4\right)
$$

$$
\leq |\mathcal{G}| \max_{g \in \mathcal{G}} \mathbb{P}\left(\left|\frac{1}{n}\sum_{i=1}^{n}\left(l_g(\boldsymbol{x}_{\sigma(i)}) - l_g(\boldsymbol{x}_{\sigma(n+i)})\right)\right| \geq \varepsilon/4\right)
$$

$$
= |\mathcal{G}| \max_{g \in \mathcal{G}} \mathbb{P}\left(\left|\frac{1}{n}\sum_{i=1}^{n}|l_g(\boldsymbol{x}_i) - l_g(\boldsymbol{x}_{n+i})|\,\epsilon_i\right| \geq \varepsilon/4\right)
$$

$$
\leq |\mathcal{G}| 2\exp\left(-\frac{\varepsilon^2 n}{32p}\right),
$$

$$(26)$$

where each $\epsilon_i$ is independently and uniformly drawn from $\{-1, 1\}$.

Noting that

$$
\|l_f - l_{f'}\|_{L^1(P_n)} = \frac{1}{2n}\sum_{i=1}^{2n}|l_f(\boldsymbol{x}_i) - l_{f'}(\boldsymbol{x}_i)|
$$

$$
= \frac{1}{2n}\sum_{i=1}^{2n}\left|\sum_{j=1}^{p}\left([x_{ij} - f_{\boldsymbol{\alpha}_j}(\boldsymbol{x}_{i,-j})]^2 - [x_{ij} - f'_{\boldsymbol{\alpha}_j}(\boldsymbol{x}_{i,-j})]^2\right)\right|
$$

$$
= \frac{1}{2n}\sum_{i=1}^{2n}\left|\sum_{j=1}^{p}(f_{\boldsymbol{\alpha}_j} - f'_{\boldsymbol{\alpha}_j})(f_{\boldsymbol{\alpha}_j} + f'_{\boldsymbol{\alpha}_j} - 2x_{ij})\right|
$$

$$
\leq \frac{1}{n}\sum_{i=1}^{2n}\sum_{j=1}^{p}|f_{\boldsymbol{\alpha}_j} - f'_{\boldsymbol{\alpha}_j}|,
$$

$$(27)$$

we can translate a cover of $\mathcal{L}_{\mathcal{F}}|_x$ into a cover of the function space $\mathcal{F}$, that is,

$$
|\mathcal{G}| = N\left(\frac{\varepsilon}{8}, \mathcal{L}_{\mathcal{F}}|_x, L^1(P_n)\right) \leq N_{2n}\left(\frac{\varepsilon}{16}, \mathcal{F}^p, L^1(P_n)\right) \leq N_{2n}\left(\frac{\varepsilon}{16p}, \mathcal{F}, L^1(P_n)\right)^p. \quad (28)
$$

For a class of functions $\mathcal{F}$, the pseudo dimension, denoted by $Pdim(\mathcal{F})$, is a natural measure of its complexity. According to [1], $Pdim(\mathcal{F})$ is the largest integer $m$ for which there exists $(x_1, \ldots, x_m, y_1, \ldots, y_m) \in \mathcal{X}^m \times \mathbb{R}^m$ such that for any $(b_1, \ldots, b_m) \in \{0, 1\}^m$ there exists $f \in \mathcal{F}$ such that $\forall i : f(x_i) > y_i \iff b_i = 1$. Using Theorem 18.4 in [1], we can give an upper bound on the covering number by $Pdim(\mathcal{F})$. Suppose that $Pdim(\mathcal{F}) = d$, we have

$$
\mathcal{N}_n\left(\varepsilon, \mathcal{F}, L^1(P_n)\right) \leq e(d+1)\left(\frac{2e}{\varepsilon}\right)^d. \quad (29)
$$

Moreover, based on Theorems 3 and 6 in [2], the pseudo dimension of ReLU feedforward neural network space is bounded as

$$
c \cdot SD\log(S/D) \leq \mathrm{Pdim}(\mathcal{F}) \leq C \cdot SD\log(S). \quad (30)
$$

Thus,

$$
\mathbb{P}(Q) \leq 4N_{2n}\left(\frac{\varepsilon}{16p}, \mathcal{F}, L^1(P_n)\right)^p \exp\left(-\frac{\varepsilon^2 n}{32p}\right)
$$

$$
\leq 4e^p\left(SD\log(S)+1\right)^p\left(\frac{32pe}{\varepsilon}\right)^{pSD\log(S)}\exp\left(-\frac{\varepsilon^2 n}{32p}\right). \quad (31)
$$

Let $p = o(\log n), \varepsilon = O\left(\frac{\log n}{n^{\frac{1}{4}}}\right)$ and $SD\log S = O(n^{\frac{1}{4}})$, it is obvious that $n > \frac{4p}{\varepsilon^2}$ and $\delta_1 = 4e^p\left(SD\log(S)+1\right)^p\left(\frac{32pe}{\varepsilon}\right)^{pSD\log(S)}\exp\left(-\frac{\varepsilon^2 n}{32p}\right) \lesssim 4n\left(n^{\frac{1}{4}}+1\right)^{\log n}\left(32en^{\frac{1}{4}}\right)^{(\log n)n^{\frac{1}{4}}}e^{-\frac{\sqrt{n}\log n}{32}} \to 0$. This completes the proof. $\qquad \square$

**Lemma A.3.** *There exists a constant $M_{\delta_2}$ such that with probability at least $1 - \delta_2$, the empirical risk function $R_n(\boldsymbol{\alpha})$ is $M_{\delta_2}\sqrt{p}$-Lipschitz.*

*Proof.* For the empirical risk:

$$
\begin{aligned}
|R_n(\boldsymbol{\alpha}) - R_n(\boldsymbol{\beta})| &= \left| \frac{1}{n} \sum_{j=1}^{p} \sum_{i=1}^{n} \left[ \left( x_{ij} - f_{\boldsymbol{\alpha}_j}(x_{i,-j}) \right)^2 - \left( x_{ij} - f_{\boldsymbol{\beta}_j}(x_{i,-j}) \right)^2 \right] \right| \\
&= \left| \frac{1}{n} \sum_{j=1}^{p} \sum_{i=1}^{n} \left( f_{\boldsymbol{\alpha}_j}(x_{i,-j}) - f_{\boldsymbol{\beta}_j}(x_{i,-j}) \right) \left( f_{\boldsymbol{\alpha}_j}(x_{i,-j}) + f_{\boldsymbol{\beta}_j}(x_{i,-j}) - 2x_{ij} \right) \right| \\
&\leq \frac{1}{n} \sum_{j=1}^{p} \sum_{i=1}^{n} \left| f_{\boldsymbol{\alpha}_j}(x_{i,-j}) - f_{\boldsymbol{\beta}_j}(x_{i,-j}) \right| \\
&\quad \left( \left| f_{\boldsymbol{\alpha}_j}(x_{i,-j}) - f_{\boldsymbol{\alpha}_j^*}(x_{i,-j}) \right| + \left| f_{\boldsymbol{\beta}_j}(x_{i,-j}) - f_{\boldsymbol{\alpha}_j^*}(x_{i,-j}) \right| + 2\left| \varepsilon_{ij} \right| \right) \\
&\leq \frac{1}{n} \sum_{j=1}^{p} \sum_{i=1}^{n} B \| \boldsymbol{\alpha}_j - \boldsymbol{\beta}_j \| \cdot (2 + 2|\varepsilon_{ij}|) \\
&= B \sum_{j=1}^{p} \| \boldsymbol{\alpha}_j - \boldsymbol{\beta}_j \| \left( 2 + \frac{2}{n} \sum_{i=1}^{n} |\varepsilon_{ij}| \right).
\end{aligned}
\tag{32}
$$

For a zero-mean sub-Gaussian random variable $\varepsilon_{ij}$, we know that there exists a constant $C_1$ such that $\mathbb{E}|\varepsilon_{ij}| \leq C_1\sigma$. Applying Markov's inequality:

$$
\mathbb{P}\left( \frac{1}{n} \sum_{i=1}^{n} |\varepsilon_{ij}| > C_{\delta_2} \right) \leq \frac{\frac{1}{n} \sum_{i=1}^{n} \mathbb{E}|\varepsilon_{ij}|}{C_{\delta_2}}.
\tag{33}
$$

Choosing $C_{\delta_2} = C_1\sigma/\delta_2$, we obtain

$$
|R_n(\boldsymbol{\alpha}) - R_n(\boldsymbol{\beta})| \leq B(2 + 2C_{\delta_2})\sqrt{p}\|\boldsymbol{\alpha} - \boldsymbol{\beta}\| = M_{\delta_2}\sqrt{p}\|\boldsymbol{\alpha} - \boldsymbol{\beta}\|,
\tag{34}
$$

with probability at least $1 - \delta_2$. $\qquad\square$

### A.3.3 Proof of Theorem 3.6

Define $\boldsymbol{\beta}_n = \operatorname{argmin}_{\boldsymbol{\alpha} \in \mathcal{H}^*} \|\hat{\boldsymbol{\alpha}}_n - \boldsymbol{\alpha}\|$. Let $L(\boldsymbol{\alpha}) = \sum_{j=1}^{p} \sum_{k=1}^{p-1} \left\| \boldsymbol{\Gamma}_j^{[:,k]} \right\| + \sum_{l<l'} |\theta_{ll'}|$ and $\lambda = \max(\lambda_1, \lambda_2)$. Since $L(\boldsymbol{\alpha})$ is Lipschitz, that is,

$$
\begin{aligned}
L(\boldsymbol{\beta}_n) - L(\hat{\boldsymbol{\alpha}}_n) &= \sum_{j=1}^{p} \sum_{k=1}^{p-1} \left\| \boldsymbol{\Gamma}_{\boldsymbol{\beta}_n j}^{[:,k]} \right\| + \sum_{l<l'} |\theta_{\boldsymbol{\beta}_n ll'}| - \sum_{j=1}^{p} \sum_{k=1}^{p-1} \left\| \boldsymbol{\Gamma}_{\hat{\boldsymbol{\alpha}}_n j}^{[:,k]} \right\| - \sum_{l<l'} |\theta_{\hat{\boldsymbol{\alpha}}_n ll'}| \\
&\leq \sum_{j=1}^{p} \sum_{k=1}^{p-1} \left\| \boldsymbol{\Gamma}_{\boldsymbol{\beta}_n j}^{[:,k]} - \boldsymbol{\Gamma}_{\hat{\boldsymbol{\alpha}}_n j}^{[:,k]} \right\| + \sum_{l<l'} |\theta_{\boldsymbol{\beta}_n ll'} - \theta_{\hat{\boldsymbol{\alpha}}_n ll'}| \\
&\leq \sqrt{p(p-1)} \left\| \boldsymbol{\Gamma}_{\boldsymbol{\beta}_n} - \boldsymbol{\Gamma}_{\hat{\boldsymbol{\alpha}}_n} \right\| + \frac{L(L+1)}{2} \|\boldsymbol{\theta}_{\boldsymbol{\beta}_n} - \boldsymbol{\theta}_{\hat{\boldsymbol{\alpha}}_n}\| \\
&\leq C\sqrt{p(p-1)} \|\boldsymbol{\beta}_n - \hat{\boldsymbol{\alpha}}_n\|.
\end{aligned}
\tag{35}
$$

Combined with Lemma A.2, we have

$$
\begin{aligned}
c_2 \|\boldsymbol{\beta}_n - \hat{\boldsymbol{\alpha}}_n\|^{\nu} = c_2 d(\hat{\boldsymbol{\alpha}}_n, \mathcal{H}^*)^{\nu} &\leq R(\hat{\boldsymbol{\alpha}}_n) - R(\boldsymbol{\beta}_n) \\
&\leq 2c_1 \frac{\log n}{n^{\frac{1}{4}}} + \lambda \left( L(\boldsymbol{\beta}_n) - L(\hat{\boldsymbol{\alpha}}_n) \right) \\
&\leq 2c_1 \frac{\log n}{n^{\frac{1}{4}}} + \lambda C\sqrt{p(p-1)} \|\boldsymbol{\beta}_n - \hat{\boldsymbol{\alpha}}_n\|,
\end{aligned}
\tag{36}
$$

holds with probability at least $1 - \delta_1$.

Applying Young's inequality,

$$
\begin{aligned}
\lambda C \sqrt{p(p-1)} \|\boldsymbol{\beta}_n - \hat{\boldsymbol{\alpha}}_n\| &\leq \frac{1}{\nu} \left( \frac{(c_2\nu)^{1/\nu}}{2} \|\boldsymbol{\beta}_n - \hat{\boldsymbol{\alpha}}_n\| \right)^{\nu} + \frac{\nu-1}{\nu} \left( \frac{2C\sqrt{p(p-1)}}{(c_2\nu)^{1/\nu}} \lambda \right)^{\nu/(\nu-1)} \\
&\leq \frac{c_2}{2} \|\boldsymbol{\beta}_n - \hat{\boldsymbol{\alpha}}_n\|^{\nu} + C_\nu \left( \lambda \sqrt{p(p-1)} \right)^{\nu/(\nu-1)},
\end{aligned}
\tag{37}
$$

yielding $\|\boldsymbol{\beta}_n - \hat{\boldsymbol{\alpha}}_n\| \leq C' \left( \left( \lambda \sqrt{p(p-1)} \right)^{\frac{\nu}{\nu-1}} + \frac{\log n}{n^{\frac{1}{4}}} \right)^{\frac{1}{\nu}}$. Let $p = o(\log n), SD \log(S) = O(n^{\frac{1}{4}}), \lambda = O(n^{-\frac{1}{8}})$. Since $\nu > 2$, $1 < \frac{\nu}{\nu-1} < 2$, there exists $c_3 > 0$ such that $d(\hat{\boldsymbol{\alpha}}_n, \mathcal{H}^*) \leq c_3 \left( \frac{\log n}{n^{\frac{1}{8}}} \right)^{\frac{1}{\nu-1}}$.

Let $K$ denote the inactive regularization components of $L$. Since $K$ is Lipschitz and $K(\phi(\boldsymbol{\alpha})) = K(\boldsymbol{\alpha})$, we have

$$
\begin{aligned}
\lambda_1 \sum_{l=1}^{L} \sum_{l' \in \mathcal{P}_l} |\hat{\theta}_{l,l'}| + \lambda_2 \sum_{j=1}^{p} \sum_{k \in \mathcal{B}_j} \left\| \hat{\boldsymbol{\Gamma}}_j^{[:,c_j(k)]} \right\| &\leq R_n(\phi(\boldsymbol{\beta}_n)) - R_n(\hat{\boldsymbol{\alpha}}_n) + \lambda[K(\phi(\boldsymbol{\beta}_n)) - K(\hat{\boldsymbol{\alpha}}_n)] \\
&\leq 2c_1 \frac{\log n}{n^{\frac{1}{4}}} + R(\phi(\boldsymbol{\beta}_n)) - R(\hat{\boldsymbol{\alpha}}_n) \\
&\quad + \lambda[K(\boldsymbol{\beta}_n) - K(\hat{\boldsymbol{\alpha}}_n)] \\
&\leq 2c_1 \frac{\log n}{n^{\frac{1}{4}}} + \lambda C \sqrt{p(p-1)} \|\boldsymbol{\beta}_n - \hat{\boldsymbol{\alpha}}_n\|.
\end{aligned}
\tag{38}
$$

Similarly, since $\nu > 2$, $0 < \frac{1}{\nu-1} < 1$, there exists $c_4 > 0$ such that $\sum_{j=1}^{p} \sum_{k \in \mathcal{B}_j} \left\| \hat{\boldsymbol{\Gamma}}_j^{[:,c_j(k)]} \right\| + \sum_{l=1}^{L} \sum_{l' \in \mathcal{P}_l} |\hat{\theta}_{l,l'}| \leq c_4 \log n \left( \frac{\log n}{n^{\frac{1}{8}}} \right)^{\frac{1}{\nu-1}}$. This completes the proof.

### A.3.4 Proof of Theorem 3.7

By Theorem 3.6 and Lemma A.3, we have that for all $(j,k) \in \bigcup_{j=1}^{p} \{j\} \times \mathcal{A}_j$, with probability $1 - \delta_1$, $\hat{\boldsymbol{\Gamma}}_j^{[:,c_j(k)]}$ and $\hat{\theta}_{C_j,C_k}$ are bounded away from zero as $n \to \infty$. Our analysis considers two connection structures: variable-level connection and group-level connection. The fundamental dependency principle requires that variables from statistically independent groups must exhibit no conditional dependence. That is, $\theta_{C_j,C_k}^* = 0$ implies $\boldsymbol{\Gamma}_j^{*[:,c_j(k)]} = 0$. Let $M(\boldsymbol{\alpha}) = \sum_{j=1}^{p} \sum_{k=1}^{p-1} \frac{\left\| \boldsymbol{\Gamma}_j^{[:,k]} \right\|}{\left\| \hat{\boldsymbol{\Gamma}}_j^{[:,k]} \right\|^{\gamma}} + \sum_{l<l'} \frac{|\theta_{ll'}|}{|\hat{\theta}_{ll'}|^{\gamma}}$ and $\zeta = \max(\zeta_1, \zeta_2)$. Thus,

$$
M(\boldsymbol{\alpha}^*) = \sum_{j=1}^{p} \sum_{k \notin \mathcal{B}_j} \frac{\left\| \boldsymbol{\Gamma}_j^{*[:,c_j(k)]} \right\|}{\left\| \hat{\boldsymbol{\Gamma}}_j^{[:,c_j(k)]} \right\|^{\gamma}} + \sum_{l=1}^{L} \sum_{l' \notin \mathcal{P}_l} \frac{|\theta_{ll'}^*|}{|\hat{\theta}_{ll'}|^{\gamma}} < \infty,
\tag{39}
$$

and

$$
c_2 d(\tilde{\boldsymbol{\alpha}}_n, \mathcal{H}^*)^{\nu} \leq 2c_1 \frac{\log n}{n^{\frac{1}{4}}} + \zeta \left( M(\boldsymbol{\alpha}^*) - M(\hat{\boldsymbol{\alpha}}_n) \right) \leq 2c_1 \frac{\log n}{n^{\frac{1}{4}}} + \zeta M(\boldsymbol{\alpha}^*).
\tag{40}
$$

Let $\zeta = O \left( n^{-\frac{\gamma}{8(\nu-1)} + \epsilon} \right)$, there exists $c_5 > 0$ such that $d(\tilde{\boldsymbol{\alpha}}_n, \mathcal{H}^*) \leq c_5 n^{\left( -\frac{\gamma}{8(\nu-1)} + \epsilon \right)/\nu} \to 0$ with probability $1 - \delta_1$. Thus, by Lemma A.1, $\tilde{\theta}_{C_j,C_k}$ and $\tilde{\boldsymbol{\Gamma}}_j^{[:,c_j(k)]}$ are bounded away from zero for all $k \in \mathcal{A}_j$ and large enough $n$.

To prove true negativity, we can separately prove that $\widetilde{\boldsymbol{\Gamma}}_j^{[:,c_j(k)]} = 0$ for all $(j,k) \in \bigcup_{j=1}^{p} \{j\} \times \mathcal{B}_j$ and $\widetilde{\theta}_{l,l'} = 0$ for all $(l,l') \in \bigcup_{l=1}^{L} \{l\} \times \mathcal{P}_l$. We establish the result by contradiction. Suppose there

exist some $j$ and $k \in \mathcal{B}_j$ with $\tilde{\mathbf{\Gamma}}_j^{[:,c_j(k)]} \neq \mathbf{0}$. Define $\boldsymbol{g}_n$ the vector obtained from $\tilde{\boldsymbol{\alpha}}_n$ by setting the $\tilde{\mathbf{\Gamma}}^{[:,c_j(k)]}$ component to 0, then we have $R_n(\tilde{\boldsymbol{\alpha}}_n) + \zeta_2 \frac{\left\|\tilde{\mathbf{\Gamma}}_j^{[:,c_j(k)]}\right\|}{\left\|\hat{\mathbf{\Gamma}}_j^{[:,c_j(k)]}\right\|^\gamma} \leq R_n(\boldsymbol{g}_n)$. By Lemma A.3, there exists $M_{\delta_2}$ such that

$$\zeta_2 \frac{\left\|\tilde{\mathbf{\Gamma}}_j^{[:,c_j(k)]}\right\|}{\left\|\hat{\mathbf{\Gamma}}_j^{[:,c_j(k)]}\right\|^\gamma} \leq R_n(\boldsymbol{g}_n) - R_n(\tilde{\boldsymbol{\alpha}}_n) \leq M_{\delta_2}\sqrt{p}\|\boldsymbol{g}_n - \tilde{\boldsymbol{\alpha}}_n\| = M_{\delta_2}\sqrt{p}\left\|\tilde{\mathbf{\Gamma}}_j^{[:,c_j(k)]}\right\|, \qquad (41)$$

with probability at least $1 - \delta_2$. Since $\tilde{\mathbf{\Gamma}}^{[:,c_j(k)]} \neq 0$, we deduce that $\zeta_2 \frac{1}{\left\|\hat{\mathbf{\Gamma}}_j^{[:,c_j(k)]}\right\|^\gamma} \leq M_{\delta_2}\sqrt{p}$. This contradicts Theorem 3.6, which proves that for $n$ large enough $\zeta_2 \frac{1}{\left\|\hat{\mathbf{\Gamma}}_j^{[:,c_j(k)]}\right\|^\gamma} \geq \zeta_2 c_4^{-\gamma}(\log n)^{\frac{-\gamma\nu}{\nu-1}} n^{\frac{\gamma}{8(\nu-1)}} \geq 2M_{\delta_2}\sqrt{p}$, with probability at least $1 - \delta_1$. Thus, by Bonferroni inequality, we have $\tilde{\mathbf{\Gamma}}^{[:,c_j(k)]} = 0$ for all $k \in \mathcal{B}_j$ with probability at least $1 - \delta_1 - \delta_2$. Similarly, it can be inferred that $\tilde{\theta}_{ll'} = 0$ for all $l' \in \mathcal{P}_l$ with probability at least $1 - \delta_1 - \delta_2$. This completes the proof.

## A.4 Computation

### A.4.1 Algorithm

---

**Algorithm 1:** Two-Stage Proximal Gradient Descent for NNBLNet

---

**Input:** Data $\{\boldsymbol{x}_i\}_{i=1}^n$, learning rate $\eta$, regularization parameters $\zeta_1 = \lambda_1, \zeta_2 = \lambda_2$, power $\gamma$,
   number of epochs $T$, tolerance $\epsilon$

**Output:** Estimated parameters $\{\boldsymbol{\theta}, \mathbf{\Gamma}_j, \{\Delta_j^{(l)}\}_{l=1}^D\}_{j=1}^p$

**Stage 1: Initial Estimation (non-adaptive)**

Initialize $\boldsymbol{\theta}, \mathbf{\Gamma}_j$, and $\Delta_j^{(l)}$ for all $j$ and $l$

**for** $i = 1$ **to** $T$ **do**

  **for** $j = 1$ **to** $p$ **do**

    Compute predictions: $\hat{x}_{i,j} = f_{\boldsymbol{\alpha}_j}(\boldsymbol{x}_{i,-j})$

    Compute gradients: $\nabla_{\mathbf{\Gamma}_j}, \nabla_{\boldsymbol{\theta}}, \nabla_{\Delta_j^{(l)}}$ for all $l$

    Gradient step:

      $\mathbf{\Gamma}_j^{\text{tmp}} \leftarrow \mathbf{\Gamma}_j - \eta\nabla_{\mathbf{\Gamma}_j}$

      $\boldsymbol{\theta}^{\text{tmp}} \leftarrow \boldsymbol{\theta} - \eta\nabla_{\boldsymbol{\theta}}$

      $\Delta_j^{(l)} \leftarrow \Delta_j^{(l)} - \eta\nabla_{\Delta_j^{(l)}}$ for all $l$

    Proximal update:

      $\mathbf{\Gamma}_j^{[:,k]} \leftarrow \left(1 - \frac{\eta\zeta_1}{\|\mathbf{\Gamma}_j^{[:,k],\text{tmp}}\|}\right)_+ \mathbf{\Gamma}_j^{[:,k],\text{tmp}}$

      $\theta_{ll'} \leftarrow \text{sign}(\theta_{ll'}^{\text{tmp}}) \cdot \max(|\theta_{ll'}^{\text{tmp}}| - \eta\zeta_2, 0)$

Store estimates $\hat{\mathbf{\Gamma}}, \hat{\boldsymbol{\theta}}$ and compute adaptive weights:

$w_{jk}^{(1)} \leftarrow \frac{1}{\|\hat{\mathbf{\Gamma}}_j^{[:,k]}\|^{\gamma}+\epsilon}, \quad w_{ll'}^{(2)} \leftarrow \frac{1}{|\hat{\theta}_{ll'}|^{\gamma}+\epsilon}$

**Stage 2: Adaptive Estimation**

Reinitialize $\mathbf{\Gamma}_j$ and $\boldsymbol{\theta}$, keep $\Delta_j^{(l)}$ from Stage 1 (or reinitialize optionally)

**for** $i = 1$ **to** $T$ **do**

  **for** $j = 1$ **to** $p$ **do**

    Compute predictions and gradients as in Stage 1

    Gradient step and update $\Delta_j^{(l)}$ as before

    Proximal update with adaptive weights:

      $\mathbf{\Gamma}_j^{[:,k]} \leftarrow \left(1 - \frac{\eta\zeta_1 w_{jk}^{(1)}}{\|\mathbf{\Gamma}_j^{[:,k],\text{tmp}}\|}\right)_+ \mathbf{\Gamma}_j^{[:,k],\text{tmp}}$

      $\theta_{ll'} \leftarrow \text{sign}(\theta_{ll'}^{\text{tmp}}) \cdot \max(|\theta_{ll'}^{\text{tmp}}| - \eta\zeta_2 w_{ll'}^{(2)}, 0)$

---

### A.4.2 Selection of the Regularization Parameters

Based on the convergence requirements specified in Theorems 3.6 and 3.7, we established the parameter configuration $\lambda_k = \zeta_k = cn^{-1/8}$ for $k = 1, 2$. Computationally, $n^{-1/8}$ yields values between 0.46 and 0.26 for sample sizes ranging from 500 to 50,000. To determine the optimal $c$, in Table 5, we evaluated F1-score performance across values $\{0.05, 0.1, 0.15, 0.2, 0.25, 0.3, 0.35, 0.4, 0.45, 0.5, 1, 1.5\}$ at different sample sizes. This theoretically derived value achieved stable performance across varying $n$ at $c = 0.35$. We therefore use $c = 0.35$ in practical implementations to maintain computational efficiency without compromising on accuracy.

Table 5: F1 score of NNBLNet with different sample sizes and values of $c$ under linear and nonlinear scenarios

| Scenario | $c$ | Sample size $n$ | | | | | |
|---|---|---|---|---|---|---|---|
| | | 500 | 1000 | 2000 | 5000 | 10000 | 20000 |
| | 0.05 | 0.659 (0.033) | 0.666 (0.026) | 0.673 (0.023) | 0.674 (0.022) | 0.670 (0.019) | 0.676 (0.017) |
| | 0.10 | 0.695 (0.032) | 0.704 (0.028) | 0.713 (0.022) | 0.716 (0.018) | 0.719 (0.016) | 0.717 (0.014) |
| | 0.15 | 0.720 (0.030) | 0.729 (0.025) | 0.731 (0.019) | 0.738 (0.017) | 0.735 (0.014) | 0.742 (0.013) |
| | 0.20 | 0.738 (0.024) | 0.749 (0.021) | 0.754 (0.017) | 0.759 (0.015) | 0.760 (0.012) | 0.758 (0.011) |
| | 0.25 | 0.756 (0.022) | 0.763 (0.017) | 0.769 (0.015) | 0.772 (0.013) | 0.771 (0.011) | 0.773 (0.010) |
| Nonlinear | 0.30 | 0.764 (0.017) | 0.767 (0.015) | 0.772 (0.013) | 0.777 (0.011) | 0.778 (0.010) | 0.780 (0.009) |
| | 0.35 | 0.769 (0.019) | 0.772 (0.014) | 0.775 (0.012) | 0.778 (0.009) | 0.776 (0.009) | 0.778 (0.008) |
| | 0.40 | 0.765 (0.020) | 0.768 (0.017) | 0.775 (0.013) | 0.777 (0.011) | 0.781 (0.008) | 0.779 (0.008) |
| | 0.45 | 0.762 (0.021) | 0.763 (0.019) | 0.767 (0.014) | 0.770 (0.013) | 0.773 (0.011) | 0.771 (0.010) |
| | 0.50 | 0.748 (0.022) | 0.755 (0.018) | 0.757 (0.015) | 0.762 (0.014) | 0.761 (0.013) | 0.765 (0.011) |
| | 1.00 | 0.708 (0.027) | 0.716 (0.023) | 0.724 (0.019) | 0.729 (0.018) | 0.728 (0.016) | 0.731 (0.014) |
| | 1.50 | 0.670 (0.036) | 0.677 (0.029) | 0.688 (0.024) | 0.695 (0.022) | 0.693 (0.018) | 0.698 (0.016) |
| | 0.05 | 0.645 (0.035) | 0.652 (0.028) | 0.660 (0.025) | 0.661 (0.023) | 0.657 (0.020) | 0.664 (0.018) |
| | 0.10 | 0.682 (0.033) | 0.691 (0.029) | 0.701 (0.023) | 0.704 (0.019) | 0.707 (0.017) | 0.705 (0.015) |
| | 0.15 | 0.707 (0.031) | 0.716 (0.026) | 0.718 (0.021) | 0.726 (0.018) | 0.723 (0.016) | 0.730 (0.014) |
| | 0.20 | 0.724 (0.026) | 0.735 (0.022) | 0.741 (0.018) | 0.747 (0.016) | 0.747 (0.014) | 0.746 (0.012) |
| | 0.25 | 0.742 (0.024) | 0.751 (0.019) | 0.757 (0.016) | 0.761 (0.014) | 0.760 (0.012) | 0.763 (0.011) |
| Linear | 0.30 | 0.751 (0.019) | 0.755 (0.016) | 0.760 (0.014) | 0.765 (0.012) | 0.766 (0.011) | 0.768 (0.010) |
| | 0.35 | 0.760 (0.021) | 0.765 (0.016) | 0.769 (0.014) | 0.772 (0.011) | 0.771 (0.010) | 0.773 (0.009) |
| | 0.40 | 0.752 (0.021) | 0.755 (0.018) | 0.760 (0.015) | 0.764 (0.012) | 0.767 (0.010) | 0.766 (0.009) |
| | 0.45 | 0.746 (0.022) | 0.749 (0.020) | 0.754 (0.015) | 0.757 (0.014) | 0.760 (0.012) | 0.759 (0.011) |
| | 0.50 | 0.732 (0.023) | 0.741 (0.019) | 0.744 (0.016) | 0.749 (0.015) | 0.748 (0.014) | 0.752 (0.012) |
| | 1.00 | 0.695 (0.028) | 0.703 (0.024) | 0.711 (0.020) | 0.716 (0.019) | 0.715 (0.017) | 0.718 (0.015) |
| | 1.50 | 0.657 (0.037) | 0.664 (0.030) | 0.675 (0.025) | 0.682 (0.023) | 0.681 (0.019) | 0.686 (0.017) |

For applications requiring finer calibration, the regularization parameters may alternatively be selected via cross-validation. Specifically, to avoid extensive grid search, we unified the four sparsification parameters as a single $\lambda_0$ based on theoretical analysis indicating their identical asymptotic order. Certainly, should that be necessary, we could alternatively assume distinct parameters for the four components and conduct a grid search. We then implemented a five-fold cross-validation procedure: datasets were partitioned into training and validation sets, where models were trained using objective functions (3) and (4) to obtain $f_{\alpha_j}(\boldsymbol{x}_{i,-j})$ for $j = 1, \cdots, p$. The validation loss $L = \sum_{j=1}^{p} \frac{1}{n} \sum_{i=1}^{n} \left(x_{i,j} - f_{\alpha_j}(\boldsymbol{x}_{i,-j})\right)^2$ was evaluated across candidate $\lambda_0$ values $(0.1, 0.15, 0.2, 0.25, 0.3, 0.35, 0.4, 0.5, 1.0)$, with the minimizer selected as optimal. Performance results are summarized in Table 6. This approach yielded slightly improved F1 scores compared to our original results but incurred higher computational complexity.

### A.4.3 Selection of the Hyperparameters

Regarding the hyperparameters of the neural network, we evaluated different configurations using two simulated datasets. Specifically, we investigated the effect of the number of hidden layers: (2, 3, 5, 8) and the number of units per hidden layer: (25, 50, 100). The F1-score results (summarized in Table 7) indicate minimal performance differences across configurations. To optimize computational efficiency while maintaining competitive performance, and in line with established practices in the field, we ultimately set the network architecture to 3 hidden layers with 50 units per layer.

Table 6: Performance of NNBLNet with the optimal $\lambda_0$ identified by five-fold cross-validation across six datasets (mean values and standard deviations for synthetic datasets).

| Dataset | Recall | Precision | F1 |
|---|---|---|---|
| Nonlinear | 0.878 (0.016) | 0.702 (0.021) | 0.776 (0.014) |
| Linear | 0.893 (0.017) | 0.695 (0.019) | 0.781 (0.013) |
| Friendship | 0.888 | 0.820 | 0.853 |
| Co-authorship | 0.732 | 0.698 | 0.715 |
| BRCA | 0.779 | 0.634 | 0.700 |
| LUAD | 0.658 | 0.545 | 0.596 |

Table 7: F1 score of NNBLNet with different hyperparameters under linear and nonlinear scenarios

| Layers | Units | Scenario Type | |
|---|---|---|---|
| | | Nonlinear | Linear |
| 2 | 25 | 0.743(0.019) | 0.739(0.018) |
| | 50 | 0.755(0.018) | 0.751(0.017) |
| | 100 | 0.748(0.018) | 0.742(0.018) |
| 3 | 25 | 0.765(0.016) | 0.760(0.015) |
| | 50 | 0.772(0.014) | 0.769(0.013) |
| | 100 | 0.768(0.016) | 0.765(0.015) |
| 5 | 25 | 0.769(0.018) | 0.765(0.017) |
| | 50 | 0.773(0.015) | 0.770(0.014) |
| | 100 | 0.769(0.017) | 0.766(0.017) |
| 8 | 25 | 0.768(0.019) | 0.764(0.018) |
| | 50 | 0.771(0.016) | 0.768(0.016) |
| | 100 | 0.765(0.018) | 0.762(0.018) |

#### A.4.4 Computation Cost

All experiments were conducted on a workstation equipped with an Intel Core i7-800H Processor, an Nvidia Tesla A40 GPU, and 64GB of RAM. Table 8 compares the computational time and peak memory usage of our method against the baselines across varying values of $n$ and $p$. To ensure a fair comparison, all methods were executed on a single CPU core. The results indicate that Fair Glasso is the most computationally efficient method by a significant margin. In contrast, NNBLNet, BGSL, and DeepGRNCS involve substantially computational costs, which escalate with the number of nodes. This higher cost is attributed to the iterative training of neural networks and, for BGSL, the additional overhead from MCMC sampling. Although NNBLNet is slower than the linear-based methods, it is faster than the other neural network-based approach, DeepGRNCS. Overall, NNBLNet demonstrates favorable scalability.

### A.5 Experiment Details

#### A.5.1 Synthetic Data Setting

We simulated a dataset with $n = 1000$ samples and $p = 100$ variables. The 100 nodes (variables) were partitioned into $L = 10$ groups (blocks), each containing 10 nodes. For each block, we generated a Barabási-Albert network structure with a maximum node degree of 4.

Connections were introduced between each pair of adjacent groups, i.e., group 1 is connected to group 2, group 3 is connected to group 4, $\cdots$, and group 9 is connected to group 10. For each such pair, node-level links were established such that the $i$-th node in the first group was connected to the $i$-th node in the second group (e.g., $x_1$ to $x_{11}$, $x_2$ to $x_{12}$, etc.).

Within each group, the features were generated using a recursive formula inspired by the Barabási-Albert network property. First, nodes were sorted by degree in descending order: $j_1, \ldots, j_{10}$. For the node $j_1$ with the largest degree, $x_{ij_1}$ was generated from a standard normal distribution. For

Table 8: Average computation time (in minutes, the first value in parentheses) and peak memory usage (MB, the second value in parentheses) of different methods across various sample sizes ($n$) and dimensions ($p$)

| $n$ | $p$ | NNBLNet | BGSL | Fair Glasso | DeepGRNCS |
|---|---|---|---|---|---|
| | 50 | (9.5, 61.8) | (6.3, 61.8) | (0.0, 30.3) | (15.7, 240.4) |
| 500 | 100 | (21.8, 307.1) | (11.0, 90.3) | (0.0, 45.1) | (49.6, 1120.7) |
| | 200 | (46.2, 709.5) | (14.5, 149.2) | (0.1, 69.1) | (100.9, 4490.8) |
| | 50 | (12.2, 158.2) | (8.1, 65.7) | (0.0, 31.4) | (20.1, 338.8) |
| 1000 | 100 | (28.0, 308.9) | (14.1, 91.1) | (0.1, 45.7) | (63.6, 1148.2) |
| | 200 | (59.2, 719.6) | (18.6, 152.3) | (0.1, 70.2) | (129.4, 4602.9) |
| | 50 | (15.6, 179.7) | (10.4, 70.1) | (0.2, 34.2) | (25.7, 399.2) |
| 2000 | 100 | (35.8, 428.4) | (18.0, 98.0) | (0.3, 52.2) | (79.0, 1652.6) |
| | 200 | (75.8, 929.2) | (23.8, 171.9) | (0.4, 89.9) | (150.0, 6397.4) |
| | 50 | (19.5, 215.8) | (13.0, 74.8) | (0.3, 36.6) | (32.2, 528.7) |
| 5000 | 100 | (44.8, 498.9) | (22.6, 109.2) | (0.5, 56.3) | (95.0, 1952.7) |
| | 200 | (94.7, 1102.5) | (29.8, 182.5) | (0.7, 101.8) | (185.0, 7598.6) |
| | 50 | (22.6, 241.0) | (15.0, 80.9) | (0.5, 38.5) | (37.2, 599.5) |
| 10000 | 100 | (51.8, 579.2) | (26.1, 114.7) | (0.8, 60.1) | (110.0, 2248.9) |
| | 200 | (109.5, 1247.9) | (34.4, 193.3) | (1.1, 109.7) | (215.0, 8799.1) |
| | 50 | (26.3, 257.6) | (16.8, 85.3) | (0.7, 39.8) | (42.4, 633.2) |
| 20000 | 100 | (57.3, 609.8) | (30.8, 127.2) | (1.0, 64.4) | (121.2, 2540.5) |
| | 200 | (123.9, 1335.2) | (39.7, 208.8) | (1.4, 117.9) | (242.2, 9008.7) |

each subsequent node $j_l$ ($l = 2, \ldots, 10$) with neighbors $N_{j_l}$ in the same group, the feature $x_{ij_l}$ was simulated as:

$$x_{ij_l} = \sum_{j_k \in N_{j_l}, k < l} f_k(x_{ij_k}) + \varepsilon_{ij_l}, \tag{42}$$

where for the nonlinear case: $f_k(\cdot)$ incorporated polynomial term $x^3$, interaction term $0.5x_1x_2$, $0.2 \exp x$, and $\sin x$, and for the linear case: $f_k(x) = z \cdot x$ with $z \sim \mathcal{N}(2, 1)$. The error term $\varepsilon_{ij_l}$ was generated from $\mathcal{N}(0, 0.01)$.

For each connected group pair (e.g., group 1 and group 2), we first generated the features of the first group (e.g., group 1) using the procedure described above. Then, for the second group (e.g., group 2), we generated each node's feature using a bi-level formulation. Specifically, we first generated a group-internal signal $x'_{ij_l} = \sum_{j_k \in N_{j_l}, k < l} f_k(x_{ij_k})$, then incorporated a signal from the connected node in the previously generated group, $g(x)$, along with a group-connection coefficient $\theta^* \sim \mathcal{N}(5, 1)$ as follows:

$$x_{ij_l} = \theta^* \left( x'_{ij_l} + g(x_{ij'}) \right) + \varepsilon_{ij_l}, \tag{43}$$

where $j'$ denotes the index of the connected node in the first group, and $g(x) = x^3$ for the nonlinear case and $g(x) = x$ for the linear case.

In the synthetic data generated by the aforementioned procedure, inter-group connections accounted for 37.0% of the total network edges. To better visualize the connectivity patterns, we quantified the network's structural properties using two metrics. The intra-group density of a group $C_k$ is defined as the proportion of observed edges among all possible edges within the group:

$$\text{IntraDensity}(C_k) = \frac{2 \cdot \sum_{i < j, \, i, j \in C_k} A_{ij}}{|C_k|(|C_k| - 1)}.$$

Similarly, the inter-group density between two distinct groups $C_k$ and $C_l$ ($k \neq l$) is given by the proportion of observed edges across the groups:

$$\text{InterDensity}(C_k, C_l) = \frac{\sum_{i \in C_k, \, j \in C_l} A_{ij}}{|C_k| \cdot |C_l|}.$$

For the synthetic network, the mean intra- and inter-group densities were 37.8% and 10.0%, respectively. These metrics help characterize the strength of connections within and between groups, providing a quantitative basis for analyzing the network structure.

For both linear and nonlinear patterns, the performance was assessed over 100 independent simulation replicates, with results for recall, precision, and F1-score summarized by their mean and standard deviation.

### A.5.2 Real-world Datasets

Table 9 summarizes the key features of the real-world datasets. Additional details are provided below.

Table 9: Summary of real-world datasets.

| Dataset | Nodes (No.) | Edges (No.) | Samples (No.) | Groups (No.) |
|---|---|---|---|---|
| Friendship | Students (311) | 1009 | Interactions (47127) | Gender (2) |
| Co-authorship | Authors (130) | 525 | Keywords (1903) | Publication type (6) |
| BRCA | mRNA (73) | 763 | Patients (1099) | Pathway (4) |
| LUAD | CNA (98) | 700 | Patients (507) | Pathway (13) |

• **Friendship**. This network dataset captures social interactions among students across nine classes at a high school in Marseille, France, recorded over five consecutive days in December 2013. Following standard practice in contact network analysis, we constructed a ground-truth graph where nodes correspond to students and weighted edges reflect aggregated interaction frequencies. In line with [25], node attributes were assigned based on gender, and the signals were generated by grouping the interactions into sets of four. The Friendship dataset is available at `http://www.sociopatterns.org/datasets/high-school-contact-and-friendship-networks/`.

• **Co-authorship**. This network dataset originates from ACM conference proceedings and includes 17,431 unique authors, 122,499 publications, and 1,903 technical keywords. We focused on a representative subset of authors, where nodes correspond to individual researchers. Demographic attributes were assigned according to authors' predominant conference categories, determined by their maximum publication frequency. The ground-truth network was constructed through co-authorship detection: edges were added between authors who co-published at least one paper. To generate network signals, we quantified authors' keyword usage patterns by calculating normalized frequencies of specific technical terms across their publications. The Co-authorship dataset is available at `https://dl.acm.org/`.

• **BRCA**. The BRCA dataset is derived from The Cancer Genome Atlas (TCGA) and comprises mRNA gene expression profiles from 1,099 breast cancer patients. We selected 73 genes involved in four key biological pathways: B Cell Receptor Complexes, Caspase Cascade, G1 And S Phases, and MMP Cytokine Connection, based on prior domain knowledge and pathway annotations from the KEGG database [16]. Nodes in the network represent these genes, and group structure is defined by their pathway annotations [42]. A biologically grounded reference network was constructed using curated interaction data from the STRING database [35], which integrates multiple evidence sources such as experimental data, co-expression, and pathway information. An undirected edge was placed between two genes if a high-confidence interaction was reported in STRING, reflecting known regulatory or functional associations. For network estimation, each patient's expression profile was treated as an input signal across the 73 genes.

• **LUAD**. The LUAD dataset is also sourced from TCGA and contains copy number alteration (CNA) profiles for 507 lung adenocarcinoma patients. We focused on 98 CNAs implicated in 13 distinct biological pathways (see Figure 3 for details), with pathway annotations obtained from the KEGG database, offering a more complex grouping structure compared to BRCA. Each CNA is treated as a node, and group membership is determined by pathway assignment. As with BRCA, the reference network was constructed using functional interaction information from the STRING database. Each patient's CNA profile was used as input for structure learning, with the goal of recovering sparse and modular dependencies.

BRCA and LUAD expression data were obtained from the R package *cdgsr*, pathway information from the KEGG database was obtained using *msigdbr*, and interaction information from STRING was

obtained via *STRINGdb*. The structures of the four benchmark networks reveal distinct connectivity patterns. Specifically, the proportion of inter-group connections is 47.9% (Friendship), 36.2% (Co-authorship), 60.8% (BRCA), and 74.4% (LUAD), while the corresponding intra- and inter-group density pairs are (37.8%, 10.0%); (2.5%, 2.0%); (11.1%, 3.2%); and (53.7%, 20.9%). These metrics collectively highlight the structural heterogeneity across the datasets.

## A.6 Sensitive Analysis

### A.6.1 Generalization Capability

To evaluate the generalization capability of the proposed method, we assessed the stability of its network estimates. We repeatedly drew 90% subsets of the data from two synthetic and four real-world datasets and re-estimated the networks 100 times. The Jaccard index between each re-estimated network and the original full-dataset network was then calculated. The resulting indices (mean and standard deviation) were as follows: Nonlinear: 0.913 (0.052), Linear: 0.932 (0.047), Friendship: 0.889 (0.061), Co-authorship: 0.921 (0.055), BRAC: 0.902 (0.058), and LUAD: 0.874 (0.064). These consistently high values indicate that the proposed method yields stable network structures across diverse datasets, confirming its strong generalization capability.

### A.6.2 Performance across Varying Sample Sizes and Group Sizes

To assess the robustness of our method to sample and group size specifications, we evaluated its performance under varying sample sizes ($n = 500, 1000, 2000, 5000$) with a fixed group size of ten, and under varying group sizes (ranging from 5-10, 5-20, and 5-30 individuals) with a fixed sample size of 1000. For groups containing 20 or 30 members, the network structure was simplified to a star module with node 1 as the hub connected to all other nodes. Based on the balanced baseline setting (with a ratio of group size 1:1), these configurations reflected increasing levels of group size imbalance, with approximate ratios of 2:1, 4:1 and 6:1, respectively. The last case represents the most pronounced disparity, where the largest group is six times larger than the smallest.

Results are summarized in Tables 10 and 11. Consistent with expectations, larger sample sizes enhanced the network recovery accuracy of all methods; nevertheless, NNBLNet consistently demonstrated superior performance. According to Table 11, NNBLNet maintained highly stable performance across moderate group size imbalances, with only a slight decrease in recall, precision, and F1 score observed under the most extreme 6:1 condition.

Table 10: Performance comparison of different methods for the nonlinear synthetic network with varying sample sizes: Mean (SD) over 100 replicates.

| $n$ | Method | Recall | Precision | F1 |
|---|---|---|---|---|
| 500 | NNBLNet | 0.861 (0.022) | 0.679 (0.027) | 0.769 (0.019) |
| | Fair Glasso | 0.756 (0.021) | 0.639 (0.026) | 0.693 (0.020) |
| | BGSL | 0.764 (0.022) | 0.631 (0.024) | 0.692 (0.019) |
| | DeepGRNCS | 0.782 (0.021) | 0.662 (0.023) | 0.717 (0.018) |
| 1000 | NNBLNet | 0.872 (0.016) | 0.693 (0.022) | 0.772 (0.014) |
| | Fair Glasso | 0.779 (0.014) | 0.653 (0.021) | 0.710 (0.017) |
| | BGSL | 0.790 (0.017) | 0.644 (0.018) | 0.709 (0.013) |
| | DeepGRNCS | 0.809 (0.016) | 0.681 (0.018) | 0.731 (0.017) |
| 2000 | NNBLNet | 0.879 (0.014) | 0.707 (0.019) | 0.775 (0.012) |
| | Fair Glasso | 0.787 (0.013) | 0.660 (0.018) | 0.717 (0.015) |
| | BGSL | 0.795 (0.015) | 0.652 (0.017) | 0.717 (0.014) |
| | DeepGRNCS | 0.822 (0.014) | 0.695 (0.017) | 0.752 (0.015) |
| 5000 | NNBLNet | 0.887 (0.011) | 0.716 (0.015) | 0.778 (0.009) |
| | Fair Glasso | 0.795 (0.011) | 0.667 (0.015) | 0.725 (0.012) |
| | BGSL | 0.802 (0.012) | 0.660 (0.014) | 0.724 (0.011) |
| | DeepGRNCS | 0.838 (0.011) | 0.708 (0.014) | 0.767 (0.012) |

### A.6.3 Performance when Group Label Misclassification

Even when genuine group labels (e.g., pathway information) are available, misclassification may still occur and impair model performance. To assess the impact of such label errors, we randomly

Table 11: Performance comparison of different methods for the nonlinear synthetic network with varying group imbalance ratios: Mean (SD) over 100 replicates.

| Ratio | Method | Recall | Precision | F1 |
|---|---|---|---|---|
| 1:1 | NNBLNet | 0.872 (0.016) | 0.693 (0.022) | 0.772 (0.014) |
| | Fair Glasso | 0.779 (0.014) | 0.653 (0.021) | 0.710 (0.017) |
| | BGSL | 0.790 (0.017) | 0.644 (0.018) | 0.709 (0.013) |
| | DeepGRNCS | 0.809(0.016) | 0.681(0.018) | 0.731(0.017) |
| 2:1 | NNBLNet | 0.874 (0.017) | 0.691 (0.023) | 0.771 (0.015) |
| | Fair Glasso | 0.773 (0.015) | 0.649 (0.022) | 0.706 (0.018) |
| | BGSL | 0.782 (0.018) | 0.640 (0.020) | 0.704 (0.015) |
| | DeepGRNCS | 0.802 (0.017) | 0.670 (0.019) | 0.730 (0.016) |
| 4:1 | NNBLNet | 0.869 (0.018) | 0.688 (0.024) | 0.769 (0.016) |
| | Fair Glasso | 0.766 (0.016) | 0.642 (0.023) | 0.701 (0.018) |
| | BGSL | 0.770 (0.019) | 0.635 (0.021) | 0.698 (0.016) |
| | DeepGRNCS | 0.812 (0.017) | 0.684 (0.019) | 0.734 (0.016) |
| 6:1 | NNBLNet | 0.854 (0.020) | 0.675 (0.026) | 0.753 (0.017) |
| | Fair Glasso | 0.755 (0.018) | 0.634 (0.024) | 0.692 (0.019) |
| | BGSL | 0.756 (0.020) | 0.627 (0.023) | 0.687 (0.017) |
| | DeepGRNCS | 0.808 (0.018) | 0.677 (0.020) | 0.727 (0.017) |

scrambled node group labels in simulated data under nonlinear setting, using misclassification rates of 10%, 20%, and 30%, with each scenario repeated 100 times. We evaluated three group-aware methods—NNBLNet, Fair Glasso, and BGSL—under these conditions. As summarized in Table 12, all three methods exhibit performance degradation as label error increases. Nonetheless, NNBLNet maintains relatively higher F1 scores and consistently outperforms both BGSL and Fair Glasso, demonstrating its robustness to moderate levels of label misclassification in real-world applications.

Table 12: Performance comparison of different methods for the nonlinear synthetic network with varying levels of group label misclassification: Mean (SD) over 100 replicates.

| Error rate | Method | Recall | Precision | F1 |
|---|---|---|---|---|
| 0% | NNBLNet | 0.872 (0.016) | 0.693 (0.022) | 0.772 (0.014) |
| | Fair Glasso | 0.779 (0.014) | 0.653 (0.021) | 0.710 (0.017) |
| | BGSL | 0.790 (0.017) | 0.644 (0.018) | 0.709 (0.013) |
| 10% | NNBLNet | 0.821 (0.022) | 0.598 (0.029) | 0.691 (0.019) |
| | Fair Glasso | 0.756 (0.023) | 0.564 (0.031) | 0.662 (0.020) |
| | BGSL | 0.742 (0.024) | 0.553 (0.032) | 0.651 (0.021) |
| 20% | NNBLNet | 0.765 (0.026) | 0.541 (0.034) | 0.633 (0.022) |
| | Fair Glasso | 0.702 (0.027) | 0.509 (0.035) | 0.602 (0.023) |
| | BGSL | 0.688 (0.028) | 0.496 (0.036) | 0.588 (0.024) |
| 30% | NNBLNet | 0.702 (0.031) | 0.489 (0.039) | 0.574 (0.026) |
| | Fair Glasso | 0.648 (0.032) | 0.455 (0.040) | 0.540 (0.027) |
| | BGSL | 0.632 (0.033) | 0.442 (0.041) | 0.526 (0.028) |

## A.7 Downstream Analysis of the LUAD dataset

Based on the low-level network inferred by NNBLNet, the top three hub genes with the highest degrees are TP53, JUN, and CD44. All three are strongly supported by existing literature in the context of lung adenocarcinoma (LUAD) and lung cancer biology. Specifically, TP53 is one of the most frequently mutated tumor suppressor genes in LUAD, with mutation rates often exceeding 40–50%. Its loss or mutation contributes to genomic instability, aggressive tumor behavior, and poor prognosis in lung cancer [3]. JUN has been identified as a gene associated with response to PD-1 blockade, suggesting its potential as a biomarker for immunotherapy efficacy in non-small cell lung cancer [37]. CD44, a recognized cell surface glycoprotein and cancer stem cell marker, is commonly used to identify stem-like subpopulations in lung cancer [12].

Based on NNBLNet analysis, the most significant pathway-level connection was observed between "MMP Cytokine Connection" and "IL4 Receptor in B Lymphocytes", with extensive gene

connectivity indicating functional cross-talk. IL-4 experimentally regulates MMP expression; for instance, it suppresses IL-1–induced MMP-3 transcription in human fibroblasts by inhibiting AP-1 promoter binding [31]. In LUAD, MMPs are markedly overexpressed and contribute to ECM remodeling, tumor invasion, and metastasis [43], while IL-4 signaling modulates immune activity and tumor–microenvironment crosstalk [17]. This supports a model wherein IL-4/IL-4R signaling influences ECM dynamics in LUAD by modulating MMP pathway activity, potentially via transcriptional regulation. As shown above, the analysis further validates that the NNBLNet method can yield biologically meaningful networks.

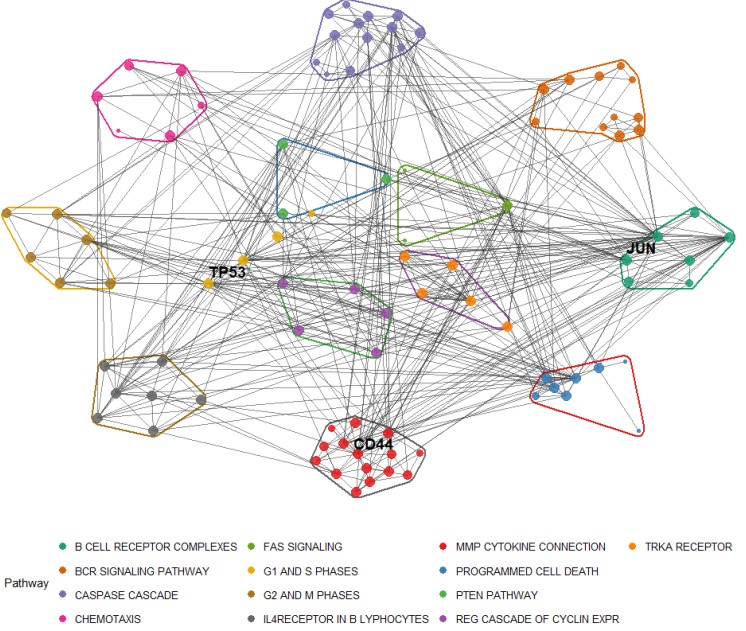

Figure 3: Network reconstruction of the LUAD dataset using NNBLNet. Nodes are color-coded by annotated pathways, and the top three hub genes are labeled.

