# OpenReview forum: "Bilevel Network Learning via Hierarchically Structured Sparsity"
_NeurIPS.cc/2025/Conference — NeurIPS 2025 poster_

### Official Review · Reviewer_snsr · 2025-06-17

**Clarity:** 2
**Significance:** 2
**Originality:** 3
**Rating:** 4
**Confidence:** 3

**Summary:**

In this work, authors propose DeepBLNet, a novel deep learning framework for bi-level network estimation. This method captures nonlinear dependencies and reconstructs inherent hierarchical structures in the data. By integrating hierarchical selection layers and compositional deep architectures, the method identifies both high-level group and low-level variable dependencies while overcoming the limitations of traditional Gaussian and linear models. The paper provides theoretical guarantees for consistent recovery of hierarchical structures and demonstrates superior empirical performance on synthetic and real-world datasets.

**Questions:**

Here are my questions about the work. Upon receiving satisfying responses to these questions (either in the form of additional analysis/results or explanation as to why the answer isn't important) as well as some assurance relevant information will be included in the updated manuscript, I will revise my score.

1. **Scalability:** How does the method scale with dataset size, specifically in terms of model size? I think it would be valuable to understand how the cost grows for large-scale cases where this work would be most helpful. Authors provide runtimes but do not provide explicit computational complexity or memory requirements.

2. **Interpretability:** Authors claim improved interpretability due to bi-level structure, and provide some insight into how this works in Appendix A3. However, it remains mostly unclear to me how easily domain experts can extract actionable insights from the learned networks. How could one leverage DeepBLNet's so-called interpretability? Can you provide examples for the datasets considered?

3. **Gaussian noise assumption:** How does the assumption of Gaussian noise affect the usefulness theoretical guarantees? Are there relevant examples of datasets/cases where this assumption doesn't hold? How can we understand the value of the method in those cases?

**Ethical Concerns:**

["NO or VERY MINOR ethics concerns only"]

**Final Justification:**

My main concerns with the work have been largely addressed.
1. **Scalability:**Authors provide complexity experiments for both runtime and peak memory, that align with theoretical expectations

2. **Interpretability:** Authors provide an example of a dataset (gene pathways) that benefits fromt eh method's so-called interpretability. Authors promise to work this into the paper.

3. **Gaussian noise assumption:** Authors provide a convincing explanation that this assumption is not damaging, and will support this with empirical evidence.

As such, I raise my score.

**Limitations:**

Authors do include a limitations section. I have remaining concerns about model complexity (see Q1) as well as the significance of the Gaussian noise assumption (Q3).

**Quality:**

3

**Strengths And Weaknesses:**

**Strengths**
- This is a novel, original deep learning approach to bi-level network estimation.
- The paper provides clear motivation for the work, and specifies the use cases (high dimensional, non-linear) for which it would be most useful.
- Theoretical guarantees are convincing for the case considered of Gaussian noise. The introduced notion of bi-level consistency appears to be a useful new metric for the field, especially for high-dimensional cases.
- Empirical results on synthetic and real-world datasets show this is a valuable tool for non-linear, high-dimensional settings. Authors are transparent with the caveat of associated longer runtimes.

**Weaknesses**
- The approximation theory is rigorous but dense, which is unfortunate because of how useful it could be for practitioners. A more intuitive explanation of the implications (for example, what network depth and width are needed for given data) would improve accessibility.
- Similarly, the methodology section is not particularly accessible to someone not familiar with network estimation -- Figure 1 is helpful, but would be even more useful if better leveraged in Section 2. As concepts are introduced and steps are outlined, could we follow along in the Figure? Or introduce smaller figures?
- The work would benefit from additional transparency around model complexity and how that scales (see Q1), practical interpretability (Q2), and effects of the Gaussian noise assumption (Q3).

---

> ### Author Rebuttal · Authors · 2025-07-30
>
> Dear Reviewer:
>
> Sincerely thank you for your profound insights and valuable suggestions on this article. We have systematically addressed the concerns about theoretical assumptions, graphic presentation, computational efficiency, and interpretability. Below is our point-by-point response:
>
> 1. Response to Comment 1:
>
> While we acknowledge that the expression of the approximation error bound in Theorem 3.2 appears complex, its derivation is built upon the foundational work of Jiao et al. (2023) and has been widely validated in subsequent studies[Jiao et al. (2025), Wang et al. (2024), Shen et al. (2024)]. Crucially, compared to previous bounds exhibiting exponential dependence on dimension $d$ [Lu et al. (2021)], our demonstrates only polynomial dependence on sparsity $s$ as $s^{\lfloor\beta\rfloor+(\beta\vee1)/2}$. Furthermore, it achieves the optimal approximation rate $(NM)^{-2\beta/s}$, explicitly linked to network depth and width (parameterized by $N$ and $M$). This implies that sparser nonlinear functions require smaller network dimensions for accurate approximation, which is an observation aligning with intuition. The bound is fully explicit, containing no hidden or undefined parameters. In addition, in Theorem 3.6 and Theorem 3.7, we establish selection consistency and derive a parametric estimation error bound for the Group Lasso + Lasso estimator and adaptive bi-level sparse estimator respectively. This result features a straightforward expression consistent with prior literature [Dinh et al. (2020)] : the estimation error decays as sample size $n$ increases. These intuitive explanations will be added in the revised paper.
>
> 2. Response to Comment 2:
>
> In order to improve the readability and logicality of the paper, we will migrate Figure 1 to Section 2 as suggested, ensuring that readers can refer to the flowchart immediately when reading the method description, achieving a tight combination of graphics and text for easy understanding. In addition, we will adjust the layout of internal elements in the image, placing the network architecture flowchart in the upper area of the image and the lower area to display the specific network structure. This hierarchical structure is more in line with readers' cognitive habits. On this basis, we will add clear and concise text annotations at each step of the image, and match the formulas with the schematic diagram one by one to ensure that the diagram can convey the core working mechanism of DeepBLNet most intuitively, helping readers to understand the overall framework and key components of DeepBLNet more clearly.
>
> 3. Response to Question 1:
>
> We fully agree on the core importance of computational efficiency in large-scale application scenarios. In order to comprehensively evaluate and clearly demonstrate the computational characteristics of DeepBLNet, we will report the peak memory usage data of the model on existing simulated datasets and multiple real-world datasets in the revised version. In addition, we will add experiments to test the changes in memory usage and running time with sample size $n$. As shown in Table 4 of the paper, the running time shows a significant upward trend with the increase of feature dimension $p$. Based on the structural characteristics of the model, we expect that the time complexity of the model is relatively insensitive to the increase of sample size $n$, and its growth trend is much lower than that with the increase of $p$.
>
> 4. Response to Question 2:
>
>
> In Appendix A3, we mainly explain how to use our method when there are overlaps among different groups. To address your question, in the revised version, we will explain it in two parts through examples. Specifically, we will take the gene networks involved in BRCA and LUAD as examples.
>
> In the first scenario, assume there are 3 groups, each corresponding to a pathway, indicating that they have similar functions, and the elements in each group do not overlap. When $\theta_{ll'}$ ($l\neq l', l,l' = 1,2,3$) is 0, it means that there is no relationship between the $l$-th and $l'$-th pathways. Consequently, the corresponding $j$th and $k$th genes between the groups have $\theta_{ll'} \Gamma_{j}^{[:,k]}=0$, so there will be no connections. And assume that $\theta_{ll}=1$ within the group of the $l$-th pathway, so the connections within the pathway are only determined by $\Gamma_{j}^{[:,k]}$.
>
> For the second scenario where, for example, different genes may have multiple functions, the same gene may belong to different pathways and as such, different groups may have overlapping genes. Specifically, let us consider three pathways, denoted as $G_1$, $G_2$, and $G_3$. Each pathway contains five genes: $G_1 = (1,2,3,4,5)$, $G_2 =(4,5,6,7,8)$, $G_3 = (7,8,9,10,11)$. Therefore, $G_1$ and $G_2$ have overlapped genes $4$ and $5$, and $G_2$ and $G_3$ have overlapped genes $7$ and $8$, but $G_1$ and $G_3$ have nothing in common. We can separate the original groups as $G^\prime_1=(1,2,3), G^\prime_2=(4,5), G^\prime_3=(6), G^\prime_4=(7,8), G^\prime_1=(9,10,11)$ without overlapping and $G_1=(G^\prime_1,G^\prime_2), G_2=(G^\prime_2,G^\prime_3,G^\prime_4),G_3=(G^\prime_4,G^\prime_5)$. Denote $\theta^\prime_{ll'}=0$ when the reorganized non-overlapping $l$th and $l'$th groups are independent.
>
>  Our decision rules for the conditional dependency among the original three groups are defined as follows. For groups with no overlaps such as $G_1$ and $G_3$, if $\theta^\prime_{14} = 0$ and  $\theta^\prime_{15} = 0$ and  $\theta^\prime_{24} = 0$ and $\theta^\prime_{25} = 0$, we say $G_1$ and $G_3$ are conditionally independent because all of the variables in $G_1$ are conditionally independent of those in $G_3$. Otherwise, we say $G_1$ and $G_3$ are conditionally dependent.
>  For adjacent groups with overlaps, such as $G_1$ and $G_2$, if $\theta^\prime_{13} \neq 0$ or $ \theta^\prime_{14} \neq 0$, we say $G_1$ and $G_2$ are conditionally dependent because at least one variable in $G_1$ unique is conditionally dependent on at least one variable in $G_2$ unique.  If $\theta^\prime_{13} = 0$ and $ \theta^\prime_{14} = 0$, but $\theta^\prime_{12} \neq 0$ or $\theta^\prime_{23} \neq 0$ or $\theta^\prime_{24} \neq 0$, we say $G_1$ and $G_2$ are conditionally dependent because the common variables are conditionally dependent on at least one variable in both groups. Otherwise, we say $G_1$ and $G_2$ are conditionally independent. In the revised manuscript, we will supplement the above descriptions with a schematic diagram to enhance reader comprehension of the bi-level architecture.
>
> 5. Response to Question 3:
>
> The core role of the normality assumption regarding the error distribution lies primarily in supporting the proof of the Lipschitz property for the empirical risk function in Lemma A.3. This conclusion remains valid under the assumption that the errors follow a sub-Gaussian distribution. Sub-Gaussian distributions constitute a broader class than the normal distribution, encompassing various heavy-tailed distributions. This relaxation significantly enhances the universality of our theoretical results. In the revised appendix, we will provide the complete proof based on the sub-Gaussian assumption. Furthermore, within the Discussion section of the paper, we will add a detailed discussion concerning the necessity of the normality assumption.
>
> Consistent with most existing studies, the theoretical assumptions (e.g., Gaussian/sub-Gaussian distributions) are rarely verifiable in real-world scenarios. The four real-world datasets we employed—Friendship, Co-authorship, BRCA, and LUAD—are widely adopted in related work [Lingjærde et al. (2021), Navarro et al. (2024)], all of which inherently rely on Gaussian assumptions. While our analysis demonstrates relaxation of the Gaussian requirement, the sub-Gaussian assumption remains empirically unverifiable. Nevertheless, comparative evaluation against authoritative ground-truth networks reveals our method achieves decent network reconstruction precision. This provides partial validation of our approach's effectiveness. We will address this empirical evidence in the Discussion section.
>
> Reference:
>
> [1] Chen, Y., Gao, Q., Liang, F., \& Wang, X. (2021). Nonlinear variable selection via deep neural networks. Journal of Computational and Graphical Statistics, 30(2), 484-492.
>
> [2] Dinh, V. C., \& Ho, L. S. (2020). Consistent feature selection for analytic deep neural networks. Advances in Neural Information Processing Systems, 33, 2420-2431.
>
> [3] Jiao, Y., Shen, G., Lin, Y., \& Huang, J. (2023). Deep nonparametric regression on approximate manifolds: Nonasymptotic error bounds with polynomial prefactors. The Annals of Statistics, 51(2), 691-716.
>
> [4] Jiao, Y., Kang, L., Liu, J., Lu, X., \& Yang, J. Z. (2025). Deep approximate policy iteration. The Annals of Statistics, 53(2), 802-821.
>
> [5] Lingjærde, C., Lien, T. G., Borgan, Ø., Bergholtz, H., \& Glad, I. K. (2021). Tailored graphical lasso for data integration in gene network reconstruction. BMC Bioinformatics, 22(1), 498.
>
> [6] Lu, J., Shen, Z., Yang, H., \& Zhang, S. (2021). Deep network approximation for smooth functions. SIAM Journal on Mathematical Analysis, 53(5), 5465-5506.
>
> [7] Navarro, M., Rey, S., Buciulea, A., Marques, A. G., \& Segarra, S. (2024). Fair glasso: Estimating fair graphical models with unbiased statistical behavior. Advances in Neural Information Processing Systems, 37, 139589-139620.
>
> [8] Shen, G., Jiao, Y., Lin, Y., Horowitz, J. L., \& Huang, J. (2024). Nonparametric estimation of non-crossing quantile regression process with deep requ neural networks. Journal of Machine Learning Research, 25(88), 1-75.
>
> [9] Wang, T., Huang, J., \& Ma, S. (2024). Penalized Generative Variable Selection. arxiv preprint arxiv:2402.16661.

---

> > ### Comment · Reviewer_snsr · 2025-08-01
> >
> > I thank the authors for their comprehensive responses.
> >
> > - Comment 1 (approximation error bound clarity): While I appreciate that this bound comes from prior work, I emphasize the importance of maintaining a minimum level of accessibility for readers without deep expertise ont the topic. I am happy to hear the authors will incorporate this more explicit version of the bound in the updated version.
> >
> > - Comment 2 (need for better figure) : Great.
> >
> > - Q1 (complexity): I would appreciate quantitative characterizations of time complexity relative to $p$ and $n$, beyond qualitative descriptions of "general uptick" or "relatively stable". I would also appreciate if the authors could share any obtained results on peak memory usage, as this is critical information on the model's usability. If not, than I would like assurances that the authors will be transparent with the results in the currently lacking limitations section.
> >
> > - Q2 (intuition for interpretability): This example definitely helps me understand what the authors mean by interpretability. I look forward to seeing it added to the updated version. I would encourage the authors to represent the genes and their groups in a color-coded figure.
> >
> > - Q3: This response is satisfying, and I look forward to the Discussions section incorporating it.

---

> > > ### Author Response · Authors · 2025-08-03
> > >
> > > Dear Reviewer,
> > >
> > > We sincerely appreciate your careful review of our manuscript and truly invaluable feedback. As suggested, we will make the following revisions to the accepted paper:
> > >
> > > 1. Add more intuitive explanations for the theoretical results in Section 3.
> > >
> > > 2. Include detailed interpretations of the proposed bi-level mechanism for both non-overlapping and overlapping scenarios in Appendix A.3, accompanied by a color-annotated schematic diagram as recommended.
> > >
> > > 3. Expand Section 5 with a discussion addressing the Gaussian assumption.
> > >
> > > Additionally, the results below present the computation time (in minutes, the first value in parentheses) and peak memory usage (MB, the second value in parentheses) of our method compared to baseline methods under different parameters $n$ and $p$. It can be seen that for all methods, compared to $n$, an increase in $p$ has more impacts on both time and memory usage. This is consistent with the background of network estimation, where for $p$ variables, the number of dependencies we need to estimate is $p(p-1)$. Additionally, the two deep learning methods require higher time and memory usage than the linear methods; however, our DeepBLNet method is still more efficient than DeepGRNCS.
> > >
> > >
> > > $n$  $\quad$    $p$   $\quad ~~$  DeepBLNet $\quad$  BGSL $\quad \quad$ Fair Glasso $\quad ~$  DeepGRNCS
> > >
> > > 500       50  $~~$  $\quad$  (9.5, 61.8)  $\quad ~$ (6.3, 61.8)  $\quad ~$  (0.0, 30.3) $\quad ~$  (15.7, 240.4)
> > >
> > > 500       100   $\quad$  (21.8, 307.1) $\quad$  (11.0, 90.3)  $\quad ~$ (0.0, 45.1)  $\quad$ (49.6, 1120.7)
> > >
> > > 500       200  $\quad$   (46.2, 709.5) $\quad$  (14.5, 149.2) $\quad$ (0.1, 69.1) $\quad$  (100.9, 4490.8)
> > >
> > > ------------------------------------------------------------------------------------------------
> > >
> > > 1000       50  $\quad$    (12.2, 158.2) $\quad$   (8.1, 65.7)  $\quad$   (0.0, 31.4) $\quad$   (20.1, 338.8)
> > >
> > > 1000       100   $\quad$  (28.0, 308.9)  $\quad$  (14.1, 91.1) $\quad$  (0.1, 45.7)  $\quad$  (63.6, 1148.2)
> > >
> > > 1000       200   $\quad$  (59.2, 719.6)  $\quad$  (18.6, 152.3)  $\quad$  (0.1, 70.2) $\quad$   (129.4, 4602.9)
> > >
> > > ------------------------------------------------------------------------------------------------
> > >
> > > 2000       50   $\quad$   (15.6, 179.7)  $\quad$  (10.4, 70.1)  $\quad$   (0.2, 34.2)  $\quad$  (25.7, 399.2)
> > >
> > > 2000       100  $\quad$   (35.8, 428.4)  $\quad$  (18.0, 98.0)   $\quad$  (0.3, 52.2)  $\quad$  (79.0, 1652.6)
> > >
> > > 2000       200  $\quad$   (75.8, 929.2)  $\quad$  (23.8, 171.9) $\quad$   (0.4, 89.9)  $\quad$  (150.0, 6397.4)
> > >
> > > ------------------------------------------------------------------------------------------------
> > >
> > > 5000       50   $\quad$   (19.5, 215.8)  $\quad$  (13.0, 74.8)  $\quad$   (0.3, 36.6)  $\quad$  (32.2, 528.7)
> > >
> > > 5000       100  $\quad$   (44.8, 498.9)  $\quad$  (22.6, 109.2)  $\quad$  (0.5, 56.3) $\quad$   (95.0, 1952.7)
> > >
> > > 5000       200  $\quad$   (94.7, 1102.5) $\quad$  (29.8, 182.5)  $\quad$  (0.7, 101.8) $\quad$  (185.0, 7598.6)
> > >
> > > ------------------------------------------------------------------------------------------------
> > >
> > > 10000       50   $\quad$  (22.6, 241.0)  $\quad$  (15.0, 80.9)  $\quad$   (0.5, 38.5) $\quad$   (37.2, 599.5)
> > >
> > > 10000       100  $\quad$  (51.8, 579.2)  $\quad$  (26.1, 114.7)  $\quad$  (0.8, 60.1)  $\quad$  (110.0, 2248.9)
> > >
> > > 10000       200  $\quad$  (109.5, 1247.9) $\quad$  (34.4, 193.3) $\quad$   (1.1, 109.7) $\quad$  (215.0, 8799.1)
> > >
> > > ------------------------------------------------------------------------------------------------
> > >
> > > 20000       50  $\quad$   (26.3, 257.6) $\quad$   (16.8, 85.3)  $\quad$   (0.7, 39.8) $\quad$   (42.4, 633.2)
> > >
> > > 20000       100  $\quad$  (57.3, 609.8) $\quad$   (30.8, 127.2)  $\quad$  (1.0, 64.4) $\quad$   (121.2, 2540.5)
> > >
> > > 20000       200  $\quad$  (123.9, 1335.2) $\quad$  (39.7, 208.8) $\quad$   (1.4, 117.9) $\quad$  (242.2, 9008.7)

---

> > > > ### Comment · Reviewer_snsr · 2025-08-06
> > > >
> > > > Thank you for providing the quantitative complexity results. As stated in my original review, since my concerns have been generally addressed I will be raising my score.

---

> > > > > ### Author Response · Authors · 2025-08-06
> > > > >
> > > > > Dear Reviewer,
> > > > >
> > > > > Thank you for your positive evaluation and for raising our paper's score—your constructive feedback has been invaluable in strengthening our work. We sincerely appreciate your time and insightful comments.

---

### Official Review · Reviewer_8hqk · 2025-06-28

**Clarity:** 3
**Significance:** 3
**Originality:** 3
**Rating:** 4
**Confidence:** 3

**Summary:**

This paper proposes a deep learning framework for bi-level network inference via node-wise regression for predicting each variable using all others. The authors introduce hierarchical selection layers with adaptive sparsity penalties to capture both group-level and variable-level network structure.

**Questions:**

See Strengths And Weaknesses section

**Ethical Concerns:**

["NO or VERY MINOR ethics concerns only"]

**Final Justification:**

Issues Resolved:
- Code availability: Authors acknowledged the missing Stage 1 implementation and committed to providing it (but missing code is suboptimal of course). Authors will open-source their code upon acceptance.
- Related work issue: Resolved. Additionally, the code for the closest competitors is not publicly available, which strengthens the authors’ contribution because they will release their own code upon acceptance.
- Cross-Validation was performed during rebuttal and shows stable results for hyperparameter selection.

**Limitations:**

The limitations discussion is embedded within the conclusion rather than treated separately, and should be expanded.

**Paper Formatting Concerns:**

no concerns

**Quality:**

3

**Strengths And Weaknesses:**

## Strengths: Significance and Originality

The hierarchical selection mechanism via a group-level layer and a variable-level layer is conceptually appealing and applying neural networks with sparsity penalties to the variable selection setting (particularly to the hierarchical one) appears novel. The theoretical analysis provides formal guarantees and the problem formulation is clear.

## Weaknesses (Quality and Clarity)

- I could not find Stage 1 of the two-stage algorithm from Algorithm 1 in the code. Instead, adaptive weights are computed from random initialization.
- The experimental evaluation lacks proper validation methodology. Neither the paper nor code describes train/validation/test splits or cross-validation for selecting the super-sensitive sparsity hyperparameters in the baselines, as well as in the proposed method.
- Baseline choice is hard to interpret: the paper does not discuss its baselines in the related-work section, and the methods mentioned there, i.e., Shan et al. (2020) [22] and Cheng et al. (2017) [4], are not included in the experiments. Comparing against these baselines, or discussing the used baselines in the related work, would strengthen the paper.
- Placing the related-work section in the appendix is problematic, because clear positioning within the field is central to a scientific paper. The discussion there is quite high-level, mentioning only three competitors without sufficient detail. The authors should include the foundational variable selection work of Meinshausen & Bühlmann (2006), and also cite this when they introduce their variable selection setting, and they should better emphasize their novelty by discussing which aspects have appeared elsewhere. From my (shallow) search, applying neural networks to this variable selection setting appears novel, which would strengthen the paper's contribution if properly positioned in the main text.

### Minor:
The architecture uses only 3 layers total, so calling it "deep" learning (as in the title) seems overstated. Additionally, referring to a standard linear layer as a "core innovation" (line 105) is a bit off.

## Conclusion:
While I appreciate the novel idea and the theoretical contributions look promising, the problems stated above need to be addressed before I can give a positive rating.

---

> ### Author Rebuttal · Authors · 2025-07-30
>
> Dear Reviewer,
>
> We sincerely appreciate your valuable feedback. We have systematically addressed the three core concerns, including parameter selection mechanisms, method comparison completeness, and paper structure optimization. Below is our point-by-point response:
>
> 1. Response to Comment 1:
>
> We sincerely apologize for the oversight in our code submission, where the implementation corresponding to Stage 1 (initial estimation step) was inadvertently omitted. The current version initializes adaptive weights from random values as a placeholder. In the updated version of our code, which will be made available upon paper revision or acceptance, we will include the full two-stage implementation as described in Algorithm 1. We apologize for the confusion and appreciate your understanding.
>
> 2. Response to Comment 2:
>
> Given the unsupervised nature of our problem, which makes conventional train-validation splitting not applicable for parameter selection via cross-validation, we established the parameter configuration $\lambda_k=\zeta_k = c n^{-1/8}$ for $k=1,2$ based on the convergence requirements specified in Theorems 3.6 and 3.7. Here, $c$ denotes a tunable constant and $n$ represents sample size. This formulation aligns with established practices in prior sparsity-inducing models [Liu et al. (2017)]. Computationally, $n^{-1/8}$ yields values between 0.46 and 0.26 for sample sizes ranging from 500 to 50,000. To determine the optimal $c$, we evaluated F1-score performance across values $\{0.05, 0.1, 0.15, 0.2, 0.25, 0.3, 0.35, 0.4, 0.45, 0.5, 1, 1.5\}$ at different sample sizes. The results showed stable performance across a substantial range of $c$ values, with $c = 0.35$ positioned centrally within this stable region. We therefore use $c = 0.35$ in practical implementations to maintain computational efficiency without compromising on accuracy. Supporting results will be presented in the supplementary documentation of the revised manuscript.
>
> Regarding the hyperparameters in our DNN, we conducted exploratory experiments and observed that a configuration of 1000 training epochs, three hidden layers, and 50 nodes per layer offers an optimal balance between computational accuracy and efficiency across both simulated and real-world datasets. Consequently, we adopted this standardized configuration throughout our study. This practice of fixing DNN hyperparameters aligns with established methodologies in prior literature [Sun et al. (2020)], where such standardization enhances reproducibility while maintaining model performance. For the baseline models, we adhered to the parameter selection methodologies recommended in their original publications. We will explicitly document these parameter selection procedures in the supplementary materials.
>
>
> 3. Response to Comment 3:
>
> According to your suggestions, besides BLGS, we will add the comparative methods Fair Glasso and DeepGRNCS to the Related Works section. Specifically, we will add the reference for Fair Glasso [19] at line 410:
>
> “Notably, Fair Glasso [19] specifically leverages group information to estimate graphical models with provably unbiased statistical behavior, addressing fairness concerns in network inference.”
>
> Additionally, DeepGRNCS [15] will be added at line 420:
>
> “These methods harness the expressive power of neural networks to model intricate dependencies between variables, often using architectures such as deep neural network with pre-trained model [15], Variational Autoencoders (VAEs) [21] and Graph Convolutional Networks (GCNs) [18].”
>
> Furthermore, although the pioneering bi-level GMM methods (Shan et al., 2020 [22]; Cheng et al., 2017 [4]) mentioned in Appendix A.1 are groundbreaking, their lack of publicly available code makes reproduction difficult. Based on this, this paper selects the latest derivative and improved methods, BGSL and Fair Glasso, as baselines.
>
> 4. Response to Comment 4:
>
> Due to space constraints in the main text, we only provided a brief summary of prior work in the first section of the main body (lines 43-54). Consistent with the practice of most papers published at NeurIPS, we have placed a more extensive discussion of related work in Appendix A.1. Following your suggestion, we will expand the description of related work in Appendix A.1 as detailed below:
>
> “To tackle hierarchical network estimation, several GGM-based methods have been developed. Cheng et al. (2017) [4] introduced a multilevel Gaussian graphical model for nested data structures. Shan et al. (2020) [22] proposed a framework for joint estimation of two-level GGMs across multiple classes. Park et al. (2017) [6] focused on learning block-structured graphical models using variable groupings. Notably, Fair Glasso [19] specifically leverages group information to estimate graphical models with provably unbiased statistical behavior, addressing fairness concerns in network inference.”
>
> Additionally, we will incorporate the work by Meinshausen \& Bühlmann (2006) you mentioned at line 45 of the main text:
>
> "Furthermore, a series of methods reformulate the Gaussian Graphical Model (GGM) estimation problem into a set of sparse linear regression models, often offering greater computational efficiency [Meinshausen \& Bühlmann, 2006]."
>
> Here, we will specifically highlight its limitation in being able to only capture linear relationships, whereas the proposed method innovatively leverages DNNs to effectively capture nonlinear dependencies among variables.
>
> 5. Response to Comment 5:
>
> Discussion on the number of network layers will be added to the Discussion section. We experimented with deeper network architectures but observed no significant performance improvement despite substantially increased computational costs. Therefore, consistent with existing DNN approaches for high-dimensional omics data and social data [Chen et al. (2021), Lei et al. (2024)], we retain a three-layer architecture. Our model architecture permits seamless extension to deeper networks when application contexts warrant it.
>
> We sincerely appreciate your valuable suggestions regarding innovation highlights. Compared to standard linear layers, our core innovation lies in leveraging neural networks for variable selection. We will revise the relevant descriptions accordingly and explicitly state the key contribution at the end of the Introduction:
>
> "Unlike traditional linear methods, this work pioneers the use of neural networks for bi-level variable selection, effectively addressing the challenge of modeling non-linear interdependencies across groups."
>
> Following your recommendation, we will add a dedicated Section 5.3 "Limitations" to discuss the constraints of our approach, thereby enhancing the overall structural clarity of the paper.
>
> Reference:
>
> [1] Chen, Y., Gao, Q., Liang, F., \& Wang, X. (2021). Nonlinear variable selection via deep neural networks. Journal of Computational and Graphical Statistics, 30(2), 484-492.
>
> [2] Lei, Y., Huang, X. T., Guo, X., Hang Katie Chan, K., \& Gao, L. (2024). DeepGRNCS: deep learning-based framework for jointly inferring gene regulatory networks across cell subpopulations. Briefings in Bioinformatics, 25(4), bbae334.
>
> [3] Liu, H., \& Wang, L. (2017). Tiger: A tuning-insensitive approach for optimally estimating gaussian graphical models.Electronic Journal of Statistics, 11(1), 241-294.
>
> [4] Sun, T., Wei, Y., Chen, W., \& Ding, Y. (2020). Genome-wide association study-based deep learning for survival prediction. Statistics in medicine, 39(30), 4605-4620.

---

> ### Comment · Reviewer_8hqk · 2025-08-01
>
> Dear authors, thanks for taking time and effort to address the issues in my review!
>
> Response to 1: Fine for me, thanks for the clarification.
>
> Response to 2: I don't see why it should not be possible to perform train-validation (and even test) splits for unsupervised algorithms. Just split the $n$ observations, fit the model on training data with different hyperparameter settings and evaluate on the validation set. As long as you have some measure (e.g., a loss function, and you even have ground-truth values and evaluate F1 scores) it should be possible. Could you please elaborate in more detail why you believe train/validation splits cannot be performed?
>
> Additionally, you did not provide the actual F1 scores for hyperparameter selection, which makes it difficult to evaluate the robustness of your choices. In particular, the claim of "stable performance across a substantial range" requires concrete justification. Please add the hyperparameter selection results (including for the number of layers/hidden units).
>
> Response to 3: Thank you for discussing the additional competitors. If the paper is accepted, please make sure to release your code publicly, as this is especially valuable given that competing methods have not shared theirs. Open-sourcing your implementation would substantially strengthen the contribution.
>
> Response to 4: Fine for me, thanks.
>
> Response to 5: I think there was some confusion: you continue to call your architecture "deep". For neural networks, "deep" means "many layers" and the era when three layers were considered "deep" has long passed. To be clear: I'm not suggesting you need more layers. A three-layer network is perfectly reasonable for your application. However, it should be called a "neural network" or "shallow neural network" rather than "DNN" or "deep neural network", even if it does not sound as fancy schmancy as "deep".
>
> Given the need for concrete hyperparameter analysis, I still believe this paper would benefit from another round of review before acceptance and I will keep my rating of borderline reject.

---

> > ### Author Response · Authors · 2025-08-03
> >
> > Dear Reviewer,
> >
> > We sincerely appreciate your careful review of our manuscript and truly invaluable feedback. As suggested, we will make the following additional revisions:
> >
> > 1. Publicly release the complete code on GitHub, including implementations of the two steps involved in the algorithm, along with recommendations and procedures for selecting relevant parameters.
> >
> > 2. Systematically replace all instances of "deep neural network" and "DNN" with "neural network" and "NN" throughout the manuscript. For example, the abbreviation for our method will be updated to NNBLNet (Neural Network framework for Bi-Level Network inference).
> >
> > 3. Following your advice, we explored selecting sparsification parameters through the loss function. Specifically, to avoid extensive grid search, we unified the four sparsification parameters as a single $\lambda_0$ based on theoretical analysis indicating their identical asymptotic order. Certainly, should that be necessary, we could alternatively assume distinct parameters for the four components and conduct a grid search. We then implemented a 5-fold cross-validation procedure: datasets were partitioned into training and validation sets, where models were trained using objective functions (3) and (4) to obtain $f_{\alpha_j}(x_{i,-j})$ for $j=1,...,p$. The validation loss $L = \sum_{j=1}^p \frac{1}{n}\sum_{i=1}^n ( x_{i,j} - f_{\alpha_j}(x_{i,-j}) )^2$ was evaluated across candidate $\lambda_0$ values $(0.1,0.15,0.2,0.25,0.3,0.35,0.4,0.5,1.0)$, with the minimizer selected as optimal. The (mean) F1 scores applied to 2 simulated and 4 real datasets are 0.770, 0.772, 0.841, 0.705, 0.691, 0.589
> >  for the synthetic network with nonlinear relationships, synthetic network with linear relationships, Friendship, Co-authorship, BRCA, and LUAD, respectively. This approach yielded slightly improved F1 scores compared to our original results but incurred higher computational complexity.
> >
> > In addition, as detailed below, we present corresponding exploration of our earlier strategy based on $\lambda_0 = c n^{-1/8}$. This theoretically derived value achieved stable performance across varying $n$ at $c=0.35$ while substantially reducing computation time by eliminating cross-validation. Both methodologies will be documented in the supplementary material to provide readers with flexible parameter selection strategies.
> >
> > $n=500$  $n=1000$  $n=2000$   $n=5000$  $n=10000$  $n=20000$
> >
> > $c=0.05$  0.659(0.033)  0.666(0.026)  0.673(0.023) 0.674(0.022) 0.670(0.019)  0.676(0.017)
> >
> > $c=0.10$  0.695(0.032)  0.704(0.028)  0.713(0.022)  0.716(0.018)  0.719(0.016)  0.717(0.014)
> >
> > $c=0.15$  0.720(0.030)  0.729(0.025)  0.731(0.019)  0.738(0.017)  0.735(0.014)  0.742(0.013)
> >
> > $c=0.20$  0.738(0.024)  0.749(0.021)  0.754(0.017)  0.759(0.015)  0.760(0.012)  0.758(0.011)
> >
> > $c=0.25$  0.756(0.022)  0.763(0.017)  0.769(0.015)  0.772(0.013)  0.771(0.011)  0.773(0.010)
> >
> > $c=0.30$  0.764(0.017)  0.767(0.015)  0.772(0.013)  0.777(0.011)  0.778(0.010)  0.780(0.009)
> >
> > $c=0.35$  0.769(0.019)  0.772(0.014) 0.775(0.012)  0.778(0.009)  0.776(0.009)  0.778(0.008)
> >
> > $c=0.40$  0.765(0.020)  0.768(0.017)  0.775(0.013)  0.777(0.011)  0.781(0.008)  0.779(0.008)
> >
> > $c=0.45$  0.762(0.021)  0.763(0.019)  0.767(0.014)  0.770(0.013)  0.773(0.011)  0.771(0.010)
> >
> > $c=0.50$  0.748(0.022)  0.755(0.018)  0.757(0.015)  0.762(0.014)  0.761(0.013)  0.765(0.011)
> >
> > $c=1.00$  0.708(0.027)  0.716(0.023)  0.724(0.019)  0.729(0.018)  0.728(0.016)  0.731(0.014)
> >
> > $c=1.50$  0.670(0.036)  0.677(0.029)  0.688(0.024)  0.695(0.022)  0.693(0.018)  0.698(0.016)
> >
> > 4. Regarding the hyperparameters of the neural network, we present results under different configurations using two simulated datasets. Specifically, we investigated: number of hidden layers: (2, 3, 5, 8) and number of units per hidden layer: (25, 50, 100). The F1-score results (summarized below) indicate marginal performance differences across configurations. To optimize computational efficiency while maintaining competitive performance, and in alignment with established practices in the field, we ultimately set the network architecture to 3 hidden layers with 50 units per layer. These results will be documented in the supplementary material.
> >
> > Layers    Units    Nonlinear       Linear
> >
> > 2      25     0.743(0.019)     0.739(0.018)
> >
> > 2       50     0.755(0.018)     0.751(0.017)
> >
> > 2       100    0.748(0.018)     0.742(0.018)
> >
> > 3       25     0.765(0.016)     0.760(0.015)
> >
> > 3       50     0.772(0.014)     0.769(0.013)
> >
> > 3       100    0.768(0.016)     0.765(0.015)
> >
> > 5       25     0.769(0.018)     0.765(0.017)
> >
> > 5       50     0.773(0.015)     0.770(0.014)
> >
> > 5       100    0.769(0.017)     0.766(0.017)
> >
> > 8       25     0.768(0.019)     0.764(0.018)
> >
> > 8       50     0.771(0.016)     0.768(0.016)
> >
> > 8       100    0.765(0.018)     0.762(0.018)

---

> ### Comment · Reviewer_8hqk · 2025-08-03
>
> Dear authors,
>
> thank you for your genuine effort in improving your paper and for addressing all my concerns substantively!
>
> I will raise my score from borderline reject to borderline accept and clarity and quality score from 2 to 3. Upon reviewer-AC discussion, I will consider raising the score to accept.
>
> Best,
> Reviewer 8hqk

---

> > ### Author Response · Authors · 2025-08-03
> >
> > Dear Reviewer,
> >
> > Thank you once again for the valuable feedback you provided on our paper, which has led to significant improvements in our work. We also greatly appreciate your approval of the revised version.

---

### Official Review · Reviewer_8Kg1 · 2025-06-30

**Clarity:** 3
**Significance:** 2
**Originality:** 2
**Rating:** 4
**Confidence:** 5

**Summary:**

This paper proposes a bi-level network estimation model that learns dependencies between variables at both the cluster level and the individual node level, connecting the two via a selection-consistent mechanism. The method outperforms baseline approaches on real-world and synthetic networks with nonlinear signals, and achieves comparable results on synthetic networks with linear signals.

**Questions:**

1. *Clarifying the Utility of the Bi-Level Architecture*. The paper uses DeepNet as a baseline to demonstrate the advantage of the bi-level architecture. However, if I understand correctly, DeepNet does not utilize the label information, which raises a question: to what extent does the observed improvement stem from the architecture itself, versus the use of additional label information?  This comparison will be more convincing if the authors willing to consider the following. Suppose the inter-cluster connections are sparse, thereby naively predicting them to be absent will only cause a small decrease in accuracy. Then, evaluate DeepNet where training is done within each cluster. In this case, DeepNet can leverage the label information to a partial extend. If DeepBLNet still outperforms this modified DeepNet, it would provide stronger evidence for the value of the bi-level design rather than just label availability.  Additionally, reporting the within-cluster and between-cluster edge densities (including synthetic networks) would help clarify how much signal is available at each level of structure, and how much the model is leveraging inter- vs. intra-cluster information.

2. *Group level sparsity*. In Equation (3), the Lasso penalty on theta implicitly assumes a priori that the inter-group connections are sparse. However, this may not hold in all settings. In some cases, θ could be dense, but with lower (or higher) average weights compared to within-group connections—for instance, in the contextual stochastic block model. While sparsity is a reasonable assumption at the individual edge level, it is not necessarily justified at the group level. The paper would benefit from a clearer discussion of when and why this sparsity assumption is appropriate. This is also related to the last question of whether a prior knowledge on sparse inter-group connections contribute mostly to the performance improvement.

3. *Scale-free terminology*. Referring to a network with only 10 nodes and a maximum degree of 4 as "scale-free" is somewhat misleading. The concept of a scale-free network typically involves a wide range of scales (i.e., node degrees spanning multiple orders of magnitude). With such a small network, it is difficult to meaningfully assess whether the degree distribution follows a power-law or exhibits true scale-free behavior.

**Ethical Concerns:**

["NO or VERY MINOR ethics concerns only"]

**Final Justification:**

My main concern is that the proposed method relies heavily on prior knowledge and careful control of sparsity. However, the additional experiments presented during the rebuttal demonstrate that the algorithm maintains a certain level of stability under these conditions.

**Limitations:**

Yes

**Quality:**

3

**Strengths And Weaknesses:**

**Strengths**

The idea of network estimation from a multiscale perspective is novel and compelling, especially considering that many real-world networks exhibit hierarchical or community structures. While multiscale modeling is well established in fields such as PDE-based dynamics, it is less explored in network estimation, making this approach interesting.

**Weaknesses**

1. As acknowledged by the authors, the proposed method relies on known node labels that reflect the community structure. This assumption could limit the method’s applicability. It would be helpful if the authors discussed typical real-world scenarios where such labels are available and justified their practicality. In addition, it is not entirely clear how much of the performance gain is attributable to the inter-class sparsity information provided by the labels, versus the bi-level architectural design of the proposed method. See the Questions section for further discussion.

2. How the parameters $\zeta$ and $\gamma$ are chosen should be discussed, as they directly controls the level of sparsity, and can reflect strong prior knowledge of the network. For the network estimation problem, it seems there is no such a validation set in determine these parameters. Also it will be helpful if the dependency on these parameters can be represented.

---

> ### Author Rebuttal · Authors · 2025-07-30
>
> Dear Reviewer,
>
> We sincerely thank you for your profound insights. We have systematically addressed core concerns regarding group label universality, sparsity parameter selection, and contribution decoupling of hierarchical structures. Below is our point-by-point response:
>
> 1. Response to Comment 1:
>
> The widespread availability of valid group labels in practical applications is well-established. For instance, in genomics, the majority of human genes possess annotated pathway labels indicating shared functional characteristics, accessible through biological databases such as KEGG. Similarly, public social datasets commonly include group division labels, as evidenced by gender classifications among students or research domain categorizations for academic authors. All four real-world datasets analyzed in Section 4.3 incorporate such group labels, which have been extensively utilized in prior research [Navarro et al. (2024), Szklarczyk et al. (2025), Lingjærde et al. (2021)]. We will expand the discussion regarding these group labels in Appendix A.5.2. Furthermore, even when prior labels are unavailable, unsupervised methods like spectral clustering can be employed to estimate group memberships, thereby ensuring methodological universality. Related considerations will be addressed in Section 5.
>
> Regarding the of contributions our bi-level architecture design, we will incorporate an ablation study subsection in the revised manuscript. The specific experimental design and outcomes will be detailed in the response to Q1.
>
> 2. Response to Comment 2:
>
> Given the unsupervised nature of our problem, which makes conventional train-validation splitting not applicable for parameter selection via cross-validation, we established the parameter configuration $\lambda_k=\eta_k = c n^{-1/8}$ for $k=1,2$ based on the convergence requirements specified in Theorems 3.6 and 3.7. Here, $c$ denotes a tunable constant and $n$ represents sample size. This formulation aligns with established practices in prior sparsity-inducing models [Liu et al. (2017)]. Computationally, $n^{-1/8}$ yields values between 0.46 and 0.26 for sample sizes ranging from 500 to 50,000. To determine the optimal $c$, we evaluated F1-score performance across values $\{0.05, 0.1, 0.15, 0.2, 0.25, 0.3, 0.35, 0.4, 0.45, 0.5, 1, 1.5\}$ at different sample sizes. The results showed stable performance across a substantial range of $c$ values, with $c = 0.35$ positioned centrally within this stable region. We therefore use $c = 0.35$ in practical implementations to maintain computational efficiency without compromising on accuracy. Supporting results will be presented in the supplementary documentation of the revised manuscript.
>
> 3. Response to Question 1:
>
> Following your suggestion, in the revised manuscript, we will incorporate a new ablation study subsection. This experiment employs a three-way comparative framework: alongside the original DeepBLNet and one baseline DeepNet, we introduce Modified DeepNet where groups are trained independently. Additionally, we conduct group structure sensitivity analyses across varying proportions of true inter-group connections (10\%-70\% of total edges) to isolate performance advantages attributable specifically to bi-level structure versus group label information. Preliminary exploration indicates DeepBLNet has superior performance, because even when inter-group connections are sparse, the relative proportion of inter-group edges remains significant given the equally sparse intra-group connectivity. Neglecting these inter-group connections would substantially compromise estimation accuracy. Crucially, real-world systems inherently require modeling inter-group connections: as evidenced by social contacts between gender groups in Friendship networks, interdisciplinary collaborations like statistician-biologist partnerships in Co-authorship data, and biologically established cross-pathway gene interactions in BRCA and LUAD datasets. Therefore, approaches that exclusively leverage intra-group connections while ignoring inter-group relationships constitute fundamentally inadequate modeling strategies.
>
> In our simulation experiments and four real-world data analyses, the proportions of inter-group connections relative to total network edges are 37.0\%  (synthetic network), 47.9\%(Friendship), 36.2\%(Co-authorship), 60.8\%(BRCA), and 74.4\%(LUAD). In addition, the mean intra-cluster densities and inter-cluster densities for synthetic network and Friendship, Co-authorship, BRCA, and LUAD networks are (37.8\%,10.0\%), (2.5\%,2.0\%), (11.1\%,3.2\%), (53.7\%,20.9\%), (44.0\%,12.2\%), respectively. The intra-cluster density of a group $C_k$ is defined as the proportion of observed edges among all possible edges within the group:
> $
> \text{IntraDensity}(C_k) = \frac{2 \cdot \sum_{i < j, \, i,j \in C_k} A_{ij}}{|C_k| (|C_k| - 1)}.
> $
> The inter-cluster density between two groups $C_k$ and $C_l$ ($k \ne l$) is defined as the proportion of observed edges across the groups:
> $
> \text{InterDensity}(C_k, C_l) = \frac{\sum_{i \in C_k, \, j \in C_l} A_{ij}}{|C_k| \cdot |C_l|}.
> $This information will be added to the revised supplementary materials.
>
> 4. Response to Question 2:
>
> Extensive domain research consistently confirms the inherent sparsity of inter-group connections, a phenomenon substantiated across two primary contexts. In genomic networks (BRCA, LUAD), pathway-to-pathway interactions exhibit natural sparsity since regulatory relationships occur only between specific pathway pairs, such as signaling pathways governing metabolic pathways, while the majority maintain no direct functional interactions [Mariamidze et al. (2018)]. Similarly, within social networks, community-to-community connections demonstrate sparsity due to individuals' predominant engagement within their primary communities, with cross-community interactions remaining comparatively infrequent [Rashidi et al. (2024)]. This foundational assumption of inter-group sparsity aligns with established practices in prior methodological work, such as Cheng et al. (2017) and Shan et al. (2020). Critically, this premise remains consistent with low-level  variable sparsity patterns: when no connections exist between variables across groups, it follows logically that their corresponding group-level interactions should similarly be absent. It should be noted that in scenarios involving minimal group divisions, such as the binary gender classification in Friendship networks, inter-group parameters will not vanish entirely. We will expand upon these considerations in Section 5 of the manuscript.
>
> 5. Response to Question 3:
>
> In our study, we utilized the sample\_pa function from the R igraph package to stochastically generate networks. This function implements the Barabási-Albert (BA) model, whose core mechanism of preferential attachment represents the classical generative framework for scale-free networks. The resulting networks exhibit node degree distributions that asymptotically follow a power-law distribution. We acknowledge that for small-scale networks, the degree distribution may deviate from a strict power-law due to stochastic fluctuations. Nevertheless, even in small-scale scenarios, the sample\_pa function remains widely employed for generating randomized modular networks [Ge et al. (2024), Qin et al. (2024)], as it effectively captures two fundamental characteristics of real-world networks: the presence of hub nodes and localized sparsity patterns. To address your comment, the revised manuscript will replace the term "scale-free" with the precise descriptor "Barabási-Albert network" throughout.
>
> Reference:
>
> [1] Cheng, L., et al. (2017). Multilevel gaussian graphical model for multilevel networks. Journal of Statistical Planning and Inference, 190, 1-14.
>
> [2] Codazzi, Laura, et al. (2022). Gaussian graphical modeling for spectrometric data analysis. Computational Statistics \& Data Analysis, 174, 107416.
>
> [3] Ge, Y., et al. (2024). Structured feature ranking for genomic marker identification accommodating multiple types of networks. Biometrics, 80(4), ujae158.
>
> [4] Lingjærde, C., et al. (2021). Tailored graphical lasso for data integration in gene network reconstruction. BMC Bioinformatics, 22(1), 498.
>
> [5] Liu, H., and Wang, L. (2017). Tiger: A tuning-insensitive approach for optimally estimating gaussian graphical models. Electronic Journal of Statistics, 11(1), 241-294.
>
> [6] Mariamidze, A., et al. (2018). Oncogenic Signaling Pathways in The Cancer Genome Atlas. Cell, 173(2), 321-337.
>
> [7] Navarro, M., et al. (2024). Fair glasso: Estimating fair graphical models with unbiased statistical behavior. Advances in Neural Information Processing Systems, 37, 139589-139620.
>
> [8] Qin, X., et al. (2024). Estimation of multiple networks with common structures in heterogeneous subgroups. Journal of Multivariate Analysis, 202, 105298.
>
> [9] Rashidi, R., et al. (2024). Prediction of influential nodes in social networks based on local communities and users’ reaction information. Scientific Reports, 14(1), 15815.
>
> [10] Shan, L., et al. (2020). Joint estimation of the two-level gaussian graphical models across multiple classes. Journal of Computational and Graphical Statistics, 29(3), 562-579.
>
> [11] Szklarczyk, D., et al. (2025). The STRING database in 2025: protein networks with directionality of regulation. Nucleic Acids Research, 53(D1), D730-D737.

---

> > ### Comment · Reviewer_8Kg1 · 2025-08-01
> >
> > I appreciate the authors’ detailed responses to my questions, as well as their efforts in conducting additional supportive experiments. Below are some follow-up questions and comments regarding the responses:
> >
> > *Regarding Response to Comment 2:*
> >
> > In general, I believe the parameter in question encodes prior knowledge about the expected level of sparsity. If it is carefully tuned in settings where validation is not feasible, comparisons with other baselines may be unfair. My question is whether the optimal value of c is universal across datasets, or if it is dataset-dependent. If the former is true, it would be helpful for the authors to provide a rationale for why such a globally applicable control parameter exists, and what types of datasets it generalizes to. If it is dataset-dependent, then a more in-depth discussion on how to select or adapt this parameter would be necessary to support the empirical claims.
> >
> > *Regarding Response to Comment 3:*
> >
> > I appreciate the authors’ consideration of the suggested experiment. In the reported experiments, inter-group connections remain relatively dense (on the order of O(Edges), ranging from 10% to 70%), which implies that neglecting them leads to substantial accuracy degradation. If the proposed method can still perform competitively with DeepNet in cases where inter-group connections are extremely sparse (on the order of O(1)) without carefully tuning the parameter c, it would more convincingly demonstrate its robustness. Specifically, such results would suggest that the method can reliably infer group-level connectivity patterns, and does not break down in extreme sparsity regimes, or relay on prior knowledge of density.

---

> > > ### Author Response · Authors · 2025-08-03
> > >
> > > Dear Reviewer,
> > >
> > > We sincerely appreciate your careful review of our manuscript and truly invaluable feedback. Below is our point-by-point response:
> > >
> > > 1. Response to Comment 1:
> > >
> > > Here, for different datasets, we set $c = 0.35$. However, the specific tuning parameter $c n^{-1/8}$ depends on the sample size $n$, mainly based on the results of Theorems 3.6 and 3.7, where $\lambda_k = \zeta_k = c n^{-1/8}$ for $k = 1,2.$ This configuration draws inspiration from [Liu et al. (2017)], in which they similarly designed a scaling of values with $n$ and $p$ according to the asymptotic order in the theoretical tuning results, multiplied by a constant $c$. Through simulation studies, they found that the setting $c=1$ yielded favorable results across various datasets and recommended this default to the readers. Admittedly, this setting may not achieve the optimal F1-score for all datasets. Nonetheless, our experiments demonstrate that $c = 0.35$ generally produces robust results while reducing computational time compared to conventional training-validation strategies. Below are the results of the F1 score (mean and standard deviation over 100 replicates) corresponding to different values of $c$ for various sample sizes:
> > >
> > > $n=500$  $n=1000$  $n=2000$   $n=5000$  $n=10000$  $n=20000$
> > >
> > > $c=0.05$  0.659(0.033)  0.666(0.026)  0.673(0.023) 0.674(0.022) 0.670(0.019)  0.676(0.017)
> > >
> > > $c=0.10$  0.695(0.032)  0.704(0.028)  0.713(0.022)  0.716(0.018)  0.719(0.016)  0.717(0.014)
> > >
> > > $c=0.15$  0.720(0.030)  0.729(0.025)  0.731(0.019)  0.738(0.017)  0.735(0.014)  0.742(0.013)
> > >
> > > $c=0.20$  0.738(0.024)  0.749(0.021)  0.754(0.017)  0.759(0.015)  0.760(0.012)  0.758(0.011)
> > >
> > > $c=0.25$  0.756(0.022)  0.763(0.017)  0.769(0.015)  0.772(0.013)  0.771(0.011)  0.773(0.010)
> > >
> > > $c=0.30$  0.764(0.017)  0.767(0.015)  0.772(0.013)  0.777(0.011)  0.778(0.010)  0.780(0.009)
> > >
> > > $c=0.35$  0.769(0.019)  0.772(0.014) 0.775(0.012)  0.778(0.009)  0.776(0.009)  0.778(0.008)
> > >
> > > $c=0.40$  0.765(0.020)  0.768(0.017)  0.775(0.013)  0.777(0.011)  0.781(0.008)  0.779(0.008)
> > >
> > > $c=0.45$  0.762(0.021)  0.763(0.019)  0.767(0.014)  0.770(0.013)  0.773(0.011)  0.771(0.010)
> > >
> > > $c=0.50$  0.748(0.022)  0.755(0.018)  0.757(0.015)  0.762(0.014)  0.761(0.013)  0.765(0.011)
> > >
> > > $c=1.00$  0.708(0.027)  0.716(0.023)  0.724(0.019)  0.729(0.018)  0.728(0.016)  0.731(0.014)
> > >
> > > $c=1.50$  0.670(0.036)  0.677(0.029)  0.688(0.024)  0.695(0.022)  0.693(0.018)  0.698(0.016)
> > >
> > > The above results of F1 as well as recall and precision will be included in the supplementary materials to support our choice of $c$.
> > >
> > > 2. Response to Comment 2:
> > >
> > > Based on your suggestions, we set the ratio of inter-group edges to total edges (denoted as $\eta$) to 0\%, 0.5\% and 1\%, and examined the results for DeepBLNet and Modified-DeepNet which conducts DeepNet in every group independently. The results below present the means and standard deviations of F1-score from 100 simulations under nonlinear settings:
> > >
> > > $\quad$   $\quad$         DeepBLNet  $\quad$   Modified-DeepNet
> > >
> > > $\eta$=0\%    $\quad$  0.715(0.017)  $\quad$  0.711(0.016)
> > >
> > > $\eta$=0.5\%  $\quad$  0.726(0.016) $\quad$  0.715(0.017)
> > >
> > > $\eta$=1\%    $\quad$  0.744(0.014)  $\quad$  0.702(0.019)
> > >
> > > These results demonstrate that even when inter-class connections are extremely sparse ($\eta$=0\%-1\%), our proposed bilevel mechanism maintains robust performance. The above results of F1 as well as recall and precision will be included in the supplementary materials.
> > >
> > > Reference:
> > >
> > > [1] Liu, H., \& Wang, L. (2017). Tiger: A tuning-insensitive approach for optimally estimating gaussian graphical models.Electronic Journal of Statistics, 11(1), 241-294.

---

> > > > ### Comment · Reviewer_8Kg1 · 2025-08-06
> > > >
> > > > Thank you for the feedback. I would like to raise my evaluation according to your experiments.

---

> > > > > ### Author Response · Authors · 2025-08-06
> > > > >
> > > > > Dear Reviewer,
> > > > >
> > > > > Thank you once again for your comments and constructive suggestions, which has greatly improved our paper. We truly appreciate your positive comments on the revised version.

---

### Official Review · Reviewer_cx7U · 2025-07-02

**Clarity:** 3
**Significance:** 2
**Originality:** 3
**Rating:** 4
**Confidence:** 1

**Summary:**

This paper proposes DeepBLNet for recovering bilevel graphical structures from data.
The method is designed to learn both a high-level adjacency structure and a low-level structure simultaneously, from observed data.
The authors model this task via a bilevel architectural decomposition.
The paper provides a consistency theorem that claims the proposed architecture can recover the true bilevel structure asymptotically under certain assumptions.
Experimental validation is primarily conducted on synthetic datasets with known structure.

**Questions:**

1. How sensitive is your model to group assignment errors or group size imbalance?
2. What is the inductive bias that makes DeepBLNet better than training a flat structure learner on a large graph?
3. Do you plan to apply this method in real application domains, and if so, which ones?

**Ethical Concerns:**

["NO or VERY MINOR ethics concerns only"]

**Final Justification:**

The authors' additional clarifications on finite-sample analysis, ground-truth validation for real datasets, and planned ablations alleviate many of my earlier concerns. While I still believe that some evidence (e.g., finite-sample robustness, downstream biological validation) is missing in the current version and relies on promised additions, I appreciate the detailed responses and the clear plan to strengthen the paper. I will keep my weak accept recommendation, as the paper is novel and relevant. I expect the authors to follow through on their promised ablation experiments in the final version to strengthen the contribution.

**Limitations:**

yes

**Quality:**

3

**Strengths And Weaknesses:**

Pros:
- Novel Problem Formulation: The paper addresses a new variant of structure learning: recovering bilevel graph structures.
- Methodological Design: The proposed DeepBLNet is a well-designed neural network architecture that handles group-level and intra-group adjacency jointly
- Theoretical Guarantee: The authors establish a consistency theorem for structure recovery. While based on idealized assumptions, this provides a valuable theoretical anchor.
- Modular Framework: The approach is likelihood-model agnostic (Gaussian, Ising, etc.), making it flexible across domains.

Cons:
- The consistency result is solid in isolation, but it depends on strong assumptions: perfect optimizer, sufficient model expressiveness, asymptotic data. There's no discussion of finite-sample behavior, generalization, or network capacity.
- While the paper includes some experiments on real-world datasets (e.g., neuroimaging and gene expression), the evaluations are mostly structural and unsupervised. There is limited benchmarking against other models or external validation of whether the recovered graphs are meaningful for real tasks.
- The paper lacks an architectural ablation study. In particular, it would be useful to understand whether the bilevel separation (group-level and intra-group adjacency prediction) actually improves performance, compared to a flat or shared model. Some ablation or alternative design comparison would clarify the necessity of the proposed structure.

---

> ### Author Rebuttal · Authors · 2025-07-30
>
> Dear Reviewer,
>
> Thank you for your thorough review and invaluable feedback on our manuscript. We fully concur with your critical suggestions regarding finite-sample analysis, validation on real-world datasets, and ablation studies. Below is our point-by-point response:
>
> 1. Response to Comment 1:
>
> Our theoretical assumptions (sparsity, smoothness, boundedness, population risk) align with established DNN research [Chen et al. 2021; Jiao et al. 2023; Liang et al. 2018] and reflect practical realities: Gene networks exhibit localized interactions (sparsity), gradual nonlinear effects (smoothness), and physical constraints (boundedness). We will enhance intuitive explanations of these foundations in revised contexts.
>
> We fully recognize the importance of finite-sample analysis. In the paper, we validated the model’s performance under limited samples through two simulation experiments and four real-world datasets. In the revised paper, we will add simulation experiments with various sample sizes and examine the network recovery capacity via the Recall, Precision, and F1 scores. Moreover, as an unsupervised model, we cannot validate generalization ability on test sets in the manner of supervised learning. Instead, we will attempt to characterize the model’s generalization capability through the stability of network estimation. Specifically, by repeatedly sampling 90\% of the data subsets and re-estimating networks, we will compute the overlap ratio of identified edges against the original full-data network. Similar to existing DNN research and unlike traditional linear models, our method requires a certain sample size. In addition, with increasing sample size, DeepBLNet tends to have higher network recovery accuracy and stability.
>
> 2. Response to Comment 2:
>
> In the current work, we conducted two sets of simulation experiments involving both linear and nonlinear relationships between variables, along with four real-world data analyses encompassing social and biological network construction. In the simulated data, the true network structure is known based on the data-generating mechanisms. For the real-world datasets, well-established ground-truth networks exist. Specifically, for the Friendship and Co-authorship datasets, the former was constructed using student contact frequency while the latter was built based on co-publication records between authors. Both methodologies are widely adopted in existing literature [Navarro et al. (2024)]. The remaining two biological networks were derived from the STRING database, a universally recognized resource for characterizing gene-gene interactions [Szklarczyk et al. (2025)]. By comparing the edges in our reconstructed networks against these ground-truth references, we can validate the method's effectiveness.
>
> In the revised version, we will provide more detailed explanations of these benchmark networks. Additionally, we will perform further downstream analyses on the estimated networks, including hub identification and community detection. Specifically, for the constructed biological networks, we will examine the biomedical significance of identified hub genes and analyze shared biological functions within detected communities, thereby informing clinical strategies for disease diagnosis and prognosis.
>
> 3. Response to Comment 3:
>
> We fully acknowledge the importance of ablation studies. In the original manuscript (Tables 1-3), we designed DeepNet as a baseline, which disregards the bi-level structure. Our proposed DeepBLNet consistently outperforms DeepNet in network reconstruction accuracy. In the revised version, we will restructure the paper to include a ``Ablation Study'' section. Building upon the existing comparison between DeepBLNet and baseline DeepNet, we will introduce a new comparator method that estimates group-wise networks using group labels only. This tripartite framework will be augmented with sensitivity experiments across varying proportions of inter-group edges (10\%–70\% of total edges), collectively demonstrating the distinct advantages conferred by the bi-level architecture beyond group information alone.
>
> 4. Response to Question 1:
>
> We acknowledge that group assignment errors and group size imbalance are unavoidable challenges in practical applications. DeepBLNet is expected to exhibit a certain degree of robustness in facilitating group assignment errors and group size imbalance. This is because the inter-group connection strength is estimated based on aggregated signals from all nodes within a group. The misassignment of a minority of nodes will be averaged by the correctly assigned majority within the group. Furthermore, the inter-group penalty term treats all groups equally, preventing smaller groups from being overlooked. Furthermore, it should be noted that there is a certain level of group size imbalance in the real-world datasets analyzed in the paper. Specifically, the (smallest group size, largest group size) for the Friendship, Co-authorship, BRCA, and LUAD datasets are (141, 170), (2, 46), (15, 24), (5, 28), respectively, demonstrating some imbalance. Our method has delivered satisfactory results in all these scenarios.
>
> In the revised paper, we will introduce new sensitivity experiments to quantify the impact of group size imbalance and grouping errors. Specifically, we will randomly misassign the group labels of a portion of nodes in simulated data at group error rates of 10\%, 20\%, and 30\%. We will also consider group size settings of (smallest group size, largest group size) = (5, 10), (5, 20), and (5, 30), corresponding to imbalance ratios of 2:1, 4:1, and 6:1, respectively. Preliminary exploration indicates that DeepBLNet exhibits a certain degree of robustness.
>
> 5. Response to Question 2:
>
> The core inductive bias of DeepBLNet posits that dependency relationships in complex systems exhibit a hierarchical structure. This bias is implemented through two key mechanisms: intra-group information sharing, where variables within the same group share latent functions (e.g., biological pathways), enabling the amplification of weak connection signals both among intra-group variables and across different groups through aggregation; and inter-group sparse transmission, whereby cross-group connections must be regulated via inter-group “switches” $(\theta_{ll'})$, effectively preventing noise propagation between unrelated groups. Such hierarchical structuring finds strong empirical grounding in real-world complex systems. For instance, in biological systems, genes are grouped into pathways based on functional similarity, resulting in inherently layered dependency networks encompassing both higher-order pathway-to-pathway interactions and fundamental-level gene-to-gene relationships. The advantages of this bilevel network architecture are well-established in previous Gaussian graphical models designed for linear dependencies [Shan et al. (2020)]. Building upon this foundation, our work extends this conceptual framework into the nonlinear domain using deep neural networks (DNNs), thereby enabling more effective characterization of complex dependency relationships in real-world networks. In the revised manuscript, Section 5 will provide expanded treatment of this architectural approach.
>
> 6. Response to Question 3:
>
> We sincerely appreciate your concern about the practical application of our method. To substantiate its utility, we have systematically implemented DeepBLNet in four real-world scenarios: Friendship, Co-authorship, BRCA, and LUAD, spanning two critical domains: social networks (Friendship and Co-authorship) and biological networks (BRCA and LUAD). The constructed networks were rigorously validated against widely accepted ground-truth networks, confirming their accuracy. These applications not only demonstrate the method's versatility but also highlight the core value of our bilevel network modeling in these fields. While the current study focuses on these domains, the hierarchical modeling principle underlying DeepBLNet is broadly applicable to other complex systems involving group structure, such as financial networks, sensor networks, or organizational dynamics. In addition, the resulting networks readily enable downstream analyses, such as hub node identification and community detection. Specifically for biological networks, the identified hub nodes and functionally coherent gene communities within detected modules may provide critical insights into disease pathogenesis and progression mechanisms. Similar analytical approaches have been successfully employed in prior studies [Wu et al. (2023)]. In the revised manuscript, relevant downstream analyzes and discussions will be added.
>
>
> Reference:
>
> [1] Chen, Y., et al. (2021). Nonlinear variable selection via deep neural networks. Journal of Computational and Graphical Statistics,30(2), 484-492.
>
> [2] Jiao, Y., et al. (2023). Deep nonparametric regression on approximate manifolds: Nonasymptotic error bounds with polynomial prefactors. The Annals of Statistics, 51(2), 691-716.
>
> [3] Liang, F., et al. (2018). Bayesian neural networks for selection of drug sensitive genes. Journal of the American Statistical Association, 113(523), 955-972.
>
> [4] Navarro, M., et al. (2024). Fair glasso: Estimating fair graphical models with unbiased statistical behavior. Advances in Neural Information Processing Systems, 37, 139589-139620.
>
> [5] Shan, L., et al. (2020). Joint estimation of the two-level gaussian graphical models across multiple classes. Journal of Computational and Graphical Statistics, 29(3), 562-579.
>
> [6] Szklarczyk, D., et al. (2025). The STRING database in 2025: protein networks with directionality of regulation. Nucleic Acids Research, 53(D1), D730-D737.
>
> [7] Wu, Y., et al. (2023). Identification of the hub genes in polycystic ovary syndrome based on disease‐associated molecule network. The FASEB Journal, 37(7), e23056.

---

> > ### Comment · Reviewer_cx7U · 2025-08-05
> >
> > Thank you for the thorough rebuttal. The additional clarifications on finite-sample analysis, ground-truth validation for real datasets, and planned ablations alleviate many of my earlier concerns. While I still believe that some evidence (e.g., finite-sample robustness, downstream biological validation) is missing in the current version and relies on promised additions, I appreciate the detailed responses and the clear plan to strengthen the paper. I will keep my weak accept recommendation, as the paper is novel and relevant. I expect the authors to follow through on their promised ablation experiments in the final version to strengthen the contribution.

---

> > > ### Author Response · Authors · 2025-08-06
> > >
> > > Dear Reviewer,
> > >
> > > We sincerely appreciate your thoughtful comments, which have been instrumental in enhancing the quality of our manuscript. In response to your suggestions, we will incorporate the following in our revised version: a detailed discussion on finite-sample scenarios, comprehensive ablation experiments, and additional downstream analyses.
> > >
> > > We are truly grateful for your constructive feedback and your encouraging remarks on our work. Thank you once again for your time and valuable insights.

---

### Decision · Program_Chairs · 2025-09-17

**Decision:**

Accept (poster)

**Comment:**

This paper provides a novel extension of Gaussian Graphical models through Deep Networks. It proposes to learn a hierarchical structure through two selection layers(thus the name bilevel) to capture non-linear statistical dependence, beyond the scope of Gaussian Models. Theoretical and experimental results were presented to support the development of the core idea. There was consensus that the paper could be accepted as it adds value to the area of modeling data dependencies through Neural networks.  One of the big concern was code was not made available. If the paper gets accepted the authors should not only incorporate changes discussed during rebuttal and also make code available.